# Sensitivity of Simulated Ammonia Fluxes in Rocky Mountain National Park to Measurement Time Resolution and Meteorological Inputs

Lillian E. Naimie<sup>1</sup>, Da Pan<sup>1</sup>, Amy P. Sullivan<sup>1</sup>, John T. Walker<sup>2</sup>, Aleksandra Djurkovic<sup>2</sup>, Bret A. Schichtel<sup>3,4</sup>, Jeffrey L. Collett, Jr.<sup>1</sup>

10 Correspondence to: Jeffrey L. Collett, Jr. (collett@colostate.edu)

Abstract. Gaseous ammonia (NH<sub>3</sub>) is an important precursor for secondary aerosol formation and contributes to reactive nitrogen deposition. NH<sub>3</sub> dry deposition is poorly quantified due to the complex bidirectional nature of NH<sub>3</sub> atmosphere-surface exchange and lack of high time-resolution in situ NH<sub>3</sub> concentration and meteorological measurements. To better quantify NH<sub>3</sub> dry deposition, measurements of NH<sub>3</sub> were made above a subalpine forest canopy in Rocky Mountain National Park (RMNP) and used with in situ micrometeorology to simulate bidirectional fluxes. NH<sub>3</sub> dry deposition was largest during the summer, with 47% of annual net NH<sub>3</sub> dry deposition occurring in June, July, and August. Because in situ, high time-resolution concentration and meteorological data are often unavailable, the impacts on estimated deposition from utilizing more commonly available biweekly NH<sub>3</sub> measurements and ERA5 meteorology were evaluated. Fluxes simulated with biweekly NH<sub>3</sub> concentrations, commonly available from NH<sub>3</sub> monitoring networks, underestimated NH<sub>3</sub> dry deposition by 45%. These fluxes were strongly correlated with 30-minute fluxes integrated to a biweekly basis (*R*<sup>2</sup> = 0.88), indicating that a correction factor could be applied to mitigate the observed bias. Application of an average NH<sub>3</sub> diel concentration pattern to the biweekly NH<sub>3</sub> concentration data removed the observed low bias. Annual NH<sub>3</sub> dry deposition from fluxes simulated with reanalysis meteorological inputs exceeded simulations using in situ meteorology measurements by a factor of 2.

### 1. Introduction

Gaseous ammonia (NH<sub>3</sub>) is an important atmospheric constituent, with effects on atmospheric chemistry and the nitrogen cycle. Atmospheric deposition of reactive nitrogen (N<sub>r</sub>) is linked to nitrogen oxides (NO<sub>x</sub>) and NH<sub>3</sub> emissions. Emissions of NO<sub>x</sub> and NH<sub>3</sub> have many potential fates, including chemical transformation, dry deposition, particle formation, and wet deposition. Anthropogenic emissions of NO<sub>x</sub> and NH<sub>3</sub> are produced predominantly by combustion and agriculture, respectively (Walker et al., 2019a), although there are also NH<sub>3</sub> emissions from traffic, wastewater treatment, and wildfires (Walker et al., 2019b; Tomsche et al., 2023). Due to increased food demand and industrialization, anthropogenic N<sub>r</sub> has been increasing

<sup>&</sup>lt;sup>1</sup>Department of Atmospheric Science, Colorado State University, Fort Collins, CO 80523, USA

<sup>&</sup>lt;sup>2</sup>United States Environmental Protection Agency, Office of Research and Development, Durham, NC 27709, USA

<sup>&</sup>lt;sup>3</sup>Cooperative Institute for Research in the Atmosphere, Colorado State University, Fort Collins, CO 80523, USA

<sup>&</sup>lt;sup>4</sup>US National Park Service, Air Resource Division, Lakewood, CO 80225-0287, USA

annually (Galloway et al., 2008; Kanakidou et al., 2016). Excess reactive nitrogen deposition has well-documented adverse effects on ecosystem health, including eutrophication, soil acidification, decreased biodiversity, and increased N in freshwater bodies (Bobbink, 1991; Baron, 2006; Holtgrieve et al., 2011; Boot et al., 2016; Zhan et al., 2017; Pan et al., 2021).

As a result of effective NO<sub>x</sub> emission controls, the balance of N<sub>r</sub> wet deposition across the US has shifted from oxidized N-dominated to reduced N-dominated, and dry deposition of NH<sub>3</sub> at times dominates total N<sub>r</sub> deposition (Li et al., 2016; Walker et al., 2019a; Driscoll et al., 2024). The National Emission Inventory (NEI) indicates that US NO<sub>x</sub> emissions were reduced by 46% between 2013 and 2023, while NH<sub>3</sub> emissions increased by 13% (US EPA, 2023).

Critical loads, deposition levels below which harmful effects are not expected to occur, have been estimated for many ecosystems (e.g., Bowman et al., 2012; Schwede and Lear, 2014). In Rocky Mountain National Park (RMNP), a critical load of 1.5 kg N ha<sup>-1</sup> yr<sup>-1</sup>, based on wet deposition of NO<sub>3</sub><sup>-</sup> and NH<sub>4</sub><sup>+</sup>, has been established to avoid adverse effects on the ecosystem (Baron, 2006). The pre-industrial nitrogen load has been estimated at 0.2 kg N ha<sup>-1</sup> yr<sup>-1</sup>, while the current wet deposition rate is as high as 3.65 kg N ha<sup>-1</sup> yr<sup>-1</sup>, approximately 15x the natural background and significantly higher than the critical load (Burns, 2003; CDPHE, 2007; Benedict et al., 2013a). Although the RMNP N<sub>r</sub> critical load only considers wet deposition of NO<sub>3</sub><sup>-</sup> and NH<sub>4</sub><sup>+</sup>, dry deposition can also contribute significantly to total N<sub>r</sub> deposition. NH<sub>3</sub> dry deposition in RMNP was estimated to be the third largest contributor to total N<sub>r</sub> deposition, accounting for 18% of N<sub>r</sub> deposition from November 2008 to November 2009 (Benedict et al., 2013a).

NH<sub>3</sub> dry deposition, however, remains a highly uncertain component of N<sub>r</sub> deposition, and fluxes are rarely measured (Walker et al., 2019b). Previous studies in RMNP have estimated NH<sub>3</sub> dry deposition using unidirectional inferential models, where the NH<sub>3</sub> deposition velocity (V<sub>d</sub>) was approximated as 70% of the HNO<sub>3</sub> deposition velocity (Beem et al., 2010; Benedict et al., 2013a; Benedict et al., 2013b) and NH<sub>3</sub> emission from the surface was ignored. In reality, NH<sub>3</sub> exchange between the atmosphere and surface is bidirectional, including deposition to and emission from the surface (Sutton et al., 1995). Several models have been developed to simulate the bidirectional exchange of NH<sub>3</sub> with the surface (Massad et al., 2010; Zhang et al., 2010; Pleim et al., 2013). Key model inputs include micrometeorology, soil and vegetation parameters, and atmospheric concentrations. In practice, fluxes can change quickly and even reverse direction with changing environmental conditions. Gaseous NH<sub>3</sub> is challenging and expensive to measure at high time resolution; lower-cost weekly or biweekly passive diffusion-based sampler measurements are more commonly utilized for long-term monitoring (Li et al., 2016; Schiferl et al., 2016; Butler et al., 2016; Hu et al., 2021). Previous efforts have used these low-cost measurements to quantify NH<sub>3</sub> dry deposition (Walker et al., 2008; Shen et al., 2016; Tanner et al., 2022). Detailed, high time-resolution meteorological observations at the location of interest are also desired when estimating dry deposition. Due to the frequent unavailability of such data, reanalysis meteorological data is often used as a substitute (Wichink Kruit et al., 2012; Schrader et al., 2018).

50

60

Schrader et al. (2018) investigated the impact of low time-resolution NH<sub>3</sub> concentrations on modeled fluxes. They found that using monthly NH<sub>3</sub> concentrations underestimates total NH<sub>3</sub> dry deposition. However, due to a linear relationship between simulations using monthly NH<sub>3</sub> concentrations and those using hourly NH<sub>3</sub> concentrations, they were able to generate a site-specific correction to compensate for the use of low time-resolution concentration data. Simulations were done using a

simplified parameterization of the bidirectional exchange model described in Massad et al. (2010), and the NH<sub>3</sub> concentrations were simulated using the LOTOS-EUROS model (Hendricks et al., 2016).

Understanding and managing these biases could unveil opportunities to estimate NH<sub>3</sub> deposition when high time-resolution, in situ concentration, and meteorological observations are unavailable. Using high time-resolution NH<sub>3</sub> concentration measurements, we provide the first estimate of NH<sub>3</sub> annual dry deposition to an RMNP forest canopy using a bidirectional exchange model driven by high time-resolution NH<sub>3</sub> concentration data and in situ micro-meteorological measurements. We use in situ data collected in RMNP to determine if site-specific correction factors suggested by Schrader et al. (2018) apply to real-world observations and whether correction factors can be employed to reduce biases associated with NH<sub>3</sub> simulations using lower-cost, low time-resolution NH<sub>3</sub> measurements such as those available from the U.S. Ammonia Monitoring network (AMoN) (Puchalski et al., 2011). We also tested if an average NH<sub>3</sub> diel pattern could be applied to reduce these biases and, if so, the length of measurements necessary to adequately describe the diel pattern. Finally, we examine biases introduced by substituting reanalysis meteorological data for high time-resolution in situ measurements.

### 2 Data and methods

### 2.1 Site location

90

Study observations were collected in RMNP in northern Colorado. The park, established to preserve the natural landscape, including montane, subalpine, and alpine ecosystems, is predominantly above 3000 m, where ecosystems developed under nutrient-deprived conditions and are therefore sensitive to excess inputs of nitrogen. Nitrogen deposition has been a historical problem in RMNP, with diatom changes documented starting in the 1950s and more recent effects, including eutrophication and changes to plant species (Baron et al., 2000; Korb and Ranker, 2001; Baron, 2006).

The area east of RMNP (Fig. 1) includes a large urban corridor and extensive agricultural activity in the plains. The Front Range urban corridor, spanning from Denver to Fort Collins, is a major source of nitrogen oxide emissions (Benedict et al., 2013b). The northeast plains of Colorado are predominantly agricultural and include major sources of NH<sub>3</sub> emissions from both animal feeding operations and crop production. The spatial pattern seen for feedlots is broadly consistent with the spatial distribution of other agricultural activities. Pan et al. (2021) found that 40% of summertime dry deposition of NH<sub>3</sub> in RMNP was associated with transport from agricultural regions to the east.

Figure 1. A map of the study region. Animal units are shown as the number of permitted animals as of 2017, scaled by an animal unit factor relative to the species. Elevation data is from the US Geological Survey Global Multi-resolution Terrain Elevation Data 2010 (GMTED2010) at 7.5-arc-second spatial resolution, or 225 m (available at: https://earthexplorer.usgs.gov/).

Data was collected at two adjacent locations for this study, both near the base of Longs Peak in Rocky Mountain National Park: a National Ecological Observatory Network (NEON) tower site (40.275903, -105.54596) and a nearby National Park Service shelter (~500 m north of the NEON tower), from September 2021 through August 2022. The study location, denoted with a star in Fig. 1, is 2750 m above sea level. The tower is surrounded by lower montane forest, comprised of predominantly evergreen needleleaf species, including ponderosa pine, juniper, and Douglas fir. There are also groves of quaking aspen located in the region. Meteorological transport to the site is generally bimodal. Prevailing downslope transport from the northwest occurs generally overnight and during the cooler months, when ammonia concentrations are typically low. The mountain-plains circulation generates daytime upslope transport, bringing air masses from the plains east of the park up into RMNP. This pattern strengthens during warmer seasons. Periods of synoptically forced sustained upslope transport are also common, especially during spring and autumn (Gebhart et al., 2011). Downslope and upslope transport patterns are not due west and east at the study site because of channelling by local topography.

At RMNP, a diel pattern in ambient NH<sub>3</sub> concentrations has commonly been observed in past measurements. This pattern is primarily driven by changes in transport patterns that carry NH<sub>3</sub> emissions to the park (Benedict et al., 2013b; Juncosa Calahorrano et al., 2024) and, sometimes, modified by changes in the atmosphere-surface exchange of NH<sub>3</sub>, especially during NH<sub>3</sub> uptake and emission from dew formation and evaporation (Wentworth et al., 2016).

# 110 2.2 Micrometeorological measurements

# 2.2.1 in situ micrometeorology

Meteorological and soil data were accessed from the RMNP NEON flux tower. The mean canopy height in the area surrounding the tower is 19 m. Temperature (mean = 6 °C), relative humidity (mean = 40%), and annual days of precipitation are highly variable at the site due to its high elevation. Mean values were calculated from September 2021 to September 2022. Snowfall typically occurs between October and May. The seasonal mean temperatures (relative humidities) are as follows: winter (December, January, and February) mean is -3 °C (30%), spring (March, April, and May) mean is 2 °C (44%), the summer (June, July, and August) mean is 15 °C (49%), and the fall (September, October, and November) mean is 8 °C (37%). Precipitation is measured at 1-minute resolution by a Belfort AEPG II 600M weighing gauge. Precipitation events were defined as periods of rainfall separated by at least one hour without precipitation. During our study period, there were 27 precipitation events in the winter, 62 in the spring, 63 in the summer, and 26 in the fall.

Meteorological data accessed from the NEON site includes wind vectors, friction velocity, Obukhov length, soil temperature, short wave radiation, relative humidity, air density, air pressure, and air temperature above the tree canopy. The meteorological observations used from the NEON tower are 30-minute mean values. Direct measurements of wind vectors, air temperature, short wave radiation, relative humidity, air density, and air pressure were used from the tower-top measurements (25 m-agl). 3D wind vectors were measured at 20 Hz using the CSAT-3 sonic anemometer (Campbell Scientific Inc., Logan, Utah, USA). Soil temperature was taken as the average across 5 collection sites within 200 m of the flux tower. Leaf area index (LAI) is estimated at the site using remotely sensed data. The square kilometer of leaf area index values surrounding the tower site is shown in Fig. S5. A mean value of 0.8 was estimated using the landscape surrounding the site. The sensitivity of simulated NH<sub>3</sub> fluxes to LAI can also be found in section 5 of the supplementary information. Additional information about each of the reported NEON datasets can be found in the Site Management and Event Reporting documentation (available at: https://doi.org/10.48443/9p2t-hi77).

NEON meteorological data contained gaps due to power outages and scheduled instrument maintenance. Across the year of data, the gaps comprised 5.8% of the data (1021 data points). To quantify the annual deposition of NH<sub>3</sub> in RMNP, these gaps were filled using the average diel pattern of fluxes during the current biweekly NH<sub>3</sub> sampling period.

# 135 2.2.2 Reanalysis meteorology data

Detailed meteorological and soil data are not available at many locations where NH<sub>3</sub> dry deposition is of interest. Reanalysis data, which combine short-range weather forecasts with assimilated observations, are a common source of meteorological data that can be used in the absence of local observations. To probe the impact of using reanalysis data in place of in situ observations, a set of bidirectional flux simulations was conducted using ERA5 hourly reanalysis data (Hersbach et al., 2020). ERA5 hourly reanalysis data has a spatial resolution of 0.25°, or approximately 31 km. The parameters used from the ERA5 data are as follows: air temperature, air pressure, dewpoint temperature, turbulent surface stress, moisture flux, sensible heat

flux, friction velocity, standard deviation of filtered subgrid orography, solar radiation, and soil temperature. Obukhov length (L) is not given in the ERA5 dataset and was calculated using Eq. (1) following Stull (1988), shown below. Obukhov Length is the characteristic length scale of turbulence in the atmospheric boundary layer and is calculated from ERA5 data using instantaneous surface sensible heat and moisture fluxes based on the suggested calculation from the European Centre for Medium-Range Weather Forecasts (Gusti, 2024).

$$L = \frac{-\overline{\theta_v'} \, u_*^3}{k \, g \, \overline{\left(w' \, \theta_v'\right)_S}},\tag{1}$$

where k is the von Karman constant, g is gravitational acceleration,  $\overline{\theta_v'}$  is the mean virtual temperature near the surface,  $\overline{w'}$   $\theta_v'$  is the surface flux of virtual potential temperature, and  $u_*$  is the friction velocity.

### 150 2.3 NH<sub>3</sub> data

# 2.3.1 Biweekly NH<sub>3</sub> measurements

Biweekly NH<sub>3</sub> ambient air concentration was measured using Radiello passive diffusion samplers purchased from Sigma Aldrich. The Radiello sampling system includes a diffusive body (part number: RAD1201) and an adsorbing cartridge (part number: RAD168) coated with phosphoric acid. NH<sub>3</sub> (g) diffuses across the exterior diffusive body and is collected on the adsorbing cartridge as ammonium (NH<sub>4</sub><sup>+</sup>) over two weeks. Collected ammonia (as NH<sub>4</sub><sup>+</sup>) is extracted from the cartridge into deionized water and analyzed on a cation IC using a 20 mM methanesulfonic acid eluent (0.5 mL min-1) on a Dionex CS12A ion exchange column with a CSRS ULTRA II suppressor and Dionex conductivity detector (Li et al., 2016). NH<sub>3</sub> passive samples were collected in duplicate ( $\sigma = \pm 0.25 \ \mu g \ m^{-3}$ ) on top of the NEON tower (25.35 m-agl). Across the study period, there were 27 sampling periods. Due to site access issues, some samples had durations longer than 2 weeks. To create a consistent dataset, all data were aggregated to a 2-week average. In the case where two samples overlapped during a 2-week period, they were combined using a weighted average. One sample was below the detection limit and was removed from this analysis. Passive NH<sub>3</sub> sampling methods have been shown to have a low bias when compared with other sampling methods, including annular denuders and Picarro Cavity Ringdown spectroscopy methods (Puchalski et al., 2011; Pan et al., 2020).

### 2.3.2 High temporal resolution NH<sub>3</sub> measurements

NH<sub>3</sub> (g) air concentration was also measured using an ion mobility spectrometer (IMS). Ion mobility spectroscopy separates ionized molecules based on their mobility through a carrier gas, under the influence of an electric field. The instrument used was the AirSentry II Point-of-Use IMS (Particle Measuring Systems, Niwot, CO). The instrument was in the National Park Service (NPS) shelter (located at 40.278129, -105.545635), 500 meters north of the NEON site, with an inlet located approximately 2 m above natural grassland. The sampling inlet was 0.635 cm Teflon tubing, heated to 40 °C to reduce NH<sub>3</sub> loss to the sampling tube. Inlet length was kept as short as possible to further prevent NH<sub>3</sub> loss. Particles were removed by a fiber filter at the tip of the inlet. Due to the high altitude of the site location, the instrument was zeroed to account for pressure

differences upon installation. Multi-point calibrations were conducted at the beginning and end of sampling. Calibration was confirmed using a known concentration ammonia gas sample split between the instrument and a phosphoric acid-coated annular denuder, where the NH<sub>3</sub> collected by the denuder is extracted into deionized water and analyzed using ion chromatography. Zero measurements were made periodically by overflowing the inlet with ultra-high purity clean air. The AirSentry samples at a 30-second frequency. During the study, the AirSentry collected 919,000 data points. The limit of detection for 30-second measurements 70 pptv. For this data analysis, NH<sub>3</sub> concentration data was averaged to 30-minute mean values. Averaging data points increases the signal-to-noise ratio. We approximate that the signal-to-noise ratio increases proportionally to the square root of the number of samples (n = 60) (Dempster, 2001). In this case, the signal-to-noise ratio increases by a factor of 7.7, reducing the limit of detection to 9 pptv for 30-minute mean NH<sub>3</sub> concentrations. Across the year of data collection, 101 points fell below the detection limit.

### 2.3.3 NH<sub>3</sub> data preparation

To investigate the effect of NH<sub>3</sub> (g) sampling time-resolution on simulated fluxes, bidirectional fluxes were simulated with concentration data at: (i) 30-minute resolution (30-minute NH<sub>3</sub>), (ii) with the 2-week integrated passive NH<sub>3</sub> (Biweekly Passive NH<sub>3</sub>), and (iii) lastly with an average diel profile applied to each day within the 2-week passive period (Average Diel Pattern NH<sub>3</sub>). The three NH<sub>3</sub> data products are shown in Fig. 2.

Figure 2. Three NH<sub>3</sub> concentration data sets are shown for the entire study period. The two-week average across each concentration data product is the same.

The 30-minute NH<sub>3</sub> concentration data is generated using a combination of data from the AirSentry NH<sub>3</sub> located at the NPS shelter and passive NH<sub>3</sub> samples collected on the NEON tower. Data gaps, due to power outages and regular maintenance, were filled using the average diel pattern across the year of data collection. Data gaps accounted for about 3000 out of more than 900,000 points across the study period. To generate a 30-minute NH<sub>3</sub> data set above the tree canopy, the data was divided

into biweekly periods that match the passive NH<sub>3</sub> collection periods. The average concentration from the AirSentry across each period was then scaled to match the biweekly passive NH<sub>3</sub> concentration. The 101 30-minute average NH<sub>3</sub> concentration values below the AirSentry detection limit, representing 0.5% of the total measurement period, were assumed to represent a random distribution below the detection limit and retained for post-process scaling from the passive observations. This preserves the temporal variability of NH<sub>3</sub> concentrations while ensuring that the average air concentration across the sampling period is consistent with the passive NH<sub>3</sub> measurements atop the NEON tower, which can differ from those above the adjacent grassland where the Air Sentry measurements are made.

The biweekly passive NH<sub>3</sub> with diel profile applied is generated using the annual average diel pattern of NH<sub>3</sub> from the AirSentry data. To determine if there are systematic differences between the NH<sub>3</sub> diel pattern at the two sites, raw and scaled AirSentry concentrations were compared to 4- and 6-hour University Research Glassware denuder measurements taken on the NEON tower. The NH<sub>3</sub> concentrations were well correlated between sites. This comparison is shown in Fig. S1. Each day of the biweekly passive period is assigned the average diel pattern, then the biweekly mean is scaled to match the biweekly passive concentration. This dataset was generated to investigate if the inclusion of a simple diel profile was sufficient to correct for the bias in bidirectional fluxes created by using low time-resolution NH<sub>3</sub> concentrations, as shown by Schrader et al. (2018). These three concentration data sets will be used for bidirectional flux simulations of NH<sub>3</sub>. For the rest of this work, the three NH<sub>3</sub> data sets will be referred to using the following nomenclature.

**30-minute NH3:** NH3 concentration data at 30-minute resolution

Biweekly NH<sub>3</sub>: Biweekly Passive NH<sub>3</sub> concentration data

Average Diel Pattern NH<sub>3</sub>: Passive NH<sub>3</sub> concentration scaled using an average diel profile from the 30-minute NH<sub>3</sub> dataset

### 2.4 Additional measurements

### 215 2.4.1 Wet deposition data

Weekly precipitation wet deposition data was obtained from the National Trends Network (NTN) (National Atmospheric Deposition Program, 2022) site at Beaver Meadows in RMNP ('CO19': located at 40.3639, -105.5810). The Beaver Meadows site location, at 2477 m elevation and located approximately 10 km north of the CASTNET site, is shown in Fig. 1.

### 2.4.2 Additional gas and particle measurements

Additional air concentration data was obtained from the U.S. EPA Clean Air Status and Trends Network (CASTNET) site at the NPS shelter ('ROM206': located at 40.278129, -105.545635). Weekly filter pack concentrations of nitric acid (HNO<sub>3</sub>) and sulfur dioxide (SO<sub>2</sub>) were used to calculate the acid ratio (Eq. 10) in the bidirectional exchange simulations of NH<sub>3</sub> (U.S. EPA, 2024a).

Weekly dry deposition of HNO<sub>3</sub>, NO<sub>3</sub>-, and NH<sub>4</sub>+ was estimated by CASTNET (US EPA, 2024b) using the weekly filter pack concentrations and historical values of deposition velocity (V<sub>d</sub>) from the U.S. EPA Multi-Layer Model (MLM) (Meyers et al., 1998). The generation of deposition velocities was discontinued in 2019. Bowker et al. (2011) found that using historical values of V<sub>d</sub> from the U.S. EPA Multi-Layer Model did not significantly bias the annual mean of deposition.

One approach to estimating  $NH_3$  deposition is to estimate the  $V_d$  as a fixed fraction (70%) of the  $V_d$  of  $HNO_3$ . This approach has been historically used to estimate the  $V_d$  of  $NH_3$  in RMNP (Beem et al., 2010; Benedict et al., 2013a; Benedict et al., 2013b).

$$V_{d}(NH_{3}) = 0.7 * V_{d}(HNO_{3}),$$
 (2)

### 2.5 Bidirectional flux modelling of NH<sub>3</sub>

230

Bidirectional NH<sub>3</sub> fluxes are simulated across the study period using the dry deposition inferential model described in Massad et al. (2010). This model was selected because it estimates both emissions and deposition of NH<sub>3</sub>, uses a compensation point framework to capture these complex dynamics, and takes into account rapidly changing micrometeorology. The simulation framework (Fig. 3) accounts for the bidirectional nature of NH<sub>3</sub> fluxes and allows for deposition and emission. The model determines if the flux will be negative (deposition) or positive (emission) based on the relationship between the atmospheric concentration ( $\chi_a$ ) at a given reference height (z) and the compensation point ( $\chi_{z0}$ ) at a defined distance (d) above the roughness length ( $\chi_{z0}$ ).

| χa               | Atmospheric ammonia concentration    | χο             | Canopy compensation point       |
|------------------|--------------------------------------|----------------|---------------------------------|
| χ <sub>z</sub> 0 | Compensation point at $(d + z_0)$    | χg             | Ground layer compensation point |
| Ra               | Aerodynamic resistance               | $\chi_{\rm s}$ | Stomata compensation point      |
| R <sub>b</sub>   | Laminar boundary layer resistance    | $f_t$          | Total flux                      |
| Rbg              | Ground boundary layer resistance     | $f_g$          | Ground flux                     |
| $R_{\rm w}$      | Cuticular resistance                 | $f_s$          | Stomatal flux                   |
| $R_{st}$         | Stomatal resistance                  | $f_{\rm w}$    | Cuticular flux                  |
| Rac              | Aerodynamic resistance in the canopy |                |                                 |

Figure 3. Dry deposition inferential model proposed in Massad et al. (2010). The table describes each model element. Arrows next to each flux show the allowed flux directions of the given pathway.

A conceptual diagram of resistances and compensation points is shown in Fig. 3. Aerodynamic ( $R_a$ ) and laminar boundary layer resistance ( $R_b$ ) capture the effects of turbulent and diffusive transfer from the atmosphere to the surface, respectively.  $R_a$  was calculated according to Thom (1975), where z is 25.35 m, d is 7.15 m, and the roughness length is 1.65 m. The stability functions  $\Psi_H$  and  $\Psi_M$  for scalars and momentum, respectively, are empirical relationships dependent on L (Thom 1975).

Displacement and roughness length were provided from the RMNP NEON Tower (NEON, 2023).

$$R_a = (k \bullet u^*)^{-1} \bullet \left(\ln\left(\frac{z-d}{z_0}\right) - \Psi_H + \Psi_M\right),\tag{3}$$

 $R_b$  is modeled as described in Xiu and Pleim (2001), where  $\gamma_{air}$  is the kinematic diffusivity of air, and  $D_{NH3}$  is the diffusivity of NH<sub>3</sub>.

$$R_b = \frac{5}{u^*} \cdot \left(\frac{\gamma_{air}}{D_{NH_3}}\right)^{2/3},\tag{4}$$

In-canopy resistance  $(R_g)$  is the sum of aerodynamic resistance within the canopy  $(R_{ac})$  and ground boundary layer resistance  $(R_{bg})$ .  $R_{ac}$  was calculated based on Nemitz et al. (2001) using Eq. (5), where  $\alpha$  is a height dependent constant calculated using Eq. 16 and Eq. 17 from Massad et al. (2010).

$$R_{ac(d+z0)} = \frac{\alpha_{(d+z0)}}{u^*} \tag{5}$$

Ground boundary layer resistance (R<sub>bg</sub>) is based on Nemitz et al. (2001), where u<sub>g</sub> is the wind speed at the ground, which we approximate as 5% of the wind speed at tower top (25 m), and z<sub>l</sub> is the upper bound height of the logarithmic wind profile above the ground, which we approximate as 10% of the canopy height(Nemitz et al., 2001).

$$R_{bg} = \left(\frac{\gamma_{air}}{p_{NH_3}} - \ln\left(\frac{p_{NH_3}}{k \cdot u_g \cdot z_l}\right)\right) \cdot \frac{1}{k \cdot u_g},\tag{6}$$

Stomata resistance (R<sub>st</sub>) captures the diffusion of NH<sub>3</sub> through plant stomata and is calculated as a minimum value related to the plant type proposed by Hicks et al. (1987). Further parameterization proposed by Nemitz et al. (2001) was used here to calculate R<sub>st</sub>, where SR (W m<sup>-2</sup>) is the solar radiation. The minimum value for R<sub>st</sub> (225 s m<sup>-1</sup>) was determined using Table 1 of Zhang et al. (2003), assuming 75% of the land surface was evergreen needleaf trees and 25% was deciduous broadleaf trees and shrubs.

$$R_{st} = \min\left\{5000 \ (s \ m^{-1}), 225 \ (s \ m^{-1}) \bullet \left(1 + \left(\frac{180}{sR}\right)\right)\right\},\tag{7}$$

Cuticular resistance (R<sub>w</sub>) was calculated according to the proposed corrected parameterization as described in Massad et al. (2010), for a forest ecosystem. When relative humidity (RH) is below 100%, Eq. (8) is used, and when RH exceeds or is equal to 100%, Eq. (9) is used.

$$R_W = 31.5 \bullet \frac{1}{AR} \bullet e^{(0.0318(100 - RH))}, \tag{8}$$

$$R_W = \frac{31.5}{AR} \,, \tag{9}$$

In both equations, AR is the acid ratio, which is calculated using the molar ratio of acids and bases in the atmosphere. The calculated acid ratio had a mean value of 1.3, a minimum of 0.22, and a maximum of 11.6. Acid ratios were the largest in the winter months.

$$AR = \frac{2 \cdot [SO_2] + [HNO_3]}{[NH_3]},\tag{10}$$

For this study period, the acid ratio was calculated using weekly CASTNET measurements of SO<sub>2</sub> and HNO<sub>3</sub> paired with our measurements of NH<sub>3</sub>.

 $\chi_{st}$  was calculated according to Massad et al. (2010). In the stomata compensation point (Eq. 11),  $\Gamma_{st}$  is the emission potential of the stomata and is approximated as 29 based on vegetation samples from the area surrounding the NEON Tower. The sampling methods and determination of this value can be found in the supplementary information. Emission potentials describe the potential for surface emission.

$$\chi_{st} = \frac{2.7457 \cdot 10^{15}}{T} \cdot e^{\left(-\frac{10378}{T}\right)} \cdot \Gamma_{st} , \qquad (11)$$

 $\chi_g$  was calculated according to Eq. (3) through Eq. (5) of Stratton et al. (2018). In Eq. (12), TAN is the concentration of total ammoniacal N (the sum of NH<sub>3</sub> and NH<sub>4</sub><sup>+</sup>) in the soil aqueous phase (mg kg<sup>-1</sup>),  $K_H$  is the Henry constant, and  $K_a$  is the equilibrium constant. TAN was estimated at 10.6 mg kg<sup>-1</sup> based on soil measurements in RMNP from Stratton et al. (2018).

NH<sub>3</sub> flux simulations are very sensitive to the TAN value. The supplementary information includes a test of the sensitivity of the flux results to TAN values within one standard deviation for the measurements taken by Stratton et al. (2018).

$$\chi_g = \frac{K_H}{1 + (10^{-pH})/(K_a)} \bullet TAN$$
, (12)

K<sub>H</sub> and K<sub>a</sub> were predicted using Eq. (13) and Eq. (14) based on the models of Montes et al. (2009), where T is temperature.

$$K_H = \left(\frac{0.2138}{T}\right) \cdot 10^{(6.123 - 1825/T)} \,, \tag{13}$$

$$K_a = 10^{\left(0.05 - \frac{2788}{T}\right)},\tag{14}$$

 $\chi_c$ , Eq. (15) below, was calculated using Eq. (12) from Massad et al. (2010).

$$\chi_{c} = \frac{\chi_{a} \cdot (R_{a} \cdot R_{b})^{-1} + \chi_{st} \cdot \left[ (R_{a} \cdot R_{st})^{-1} + (R_{b} \cdot R_{st})^{-1} + (R_{g} \cdot R_{st})^{-1} \right] + \chi_{g} \cdot (R_{b} \cdot R_{g})^{-1}}{(R_{a} \cdot R_{b})^{-1} + (R_{a} \cdot R_{st})^{-1} + (R_{b} \cdot R_{g})^{-1} + (R_{b} \cdot R_{st})^{-1} + (R_{g} \cdot R_{st})^{-1} + (R_{g} \cdot R_{st})^{-1}} , \qquad (15)$$

Compensation point at the displacement height (d) above the roughness length ( $z_0$ ) is calculated using Eq. (16) below as proposed in Massad et al. (2010).  $\chi_{zo}$  takes all other compensation points and resistances into account.

$$\chi_{z0} = \frac{\left(\frac{\chi_{a}}{R_{a}} + \frac{\chi_{g}}{R_{g}} + \frac{\chi_{c}}{R_{b}}\right)}{\left(\frac{1}{R_{a}} + \frac{1}{R_{g}} + \frac{1}{R_{b}}\right)},\tag{16}$$

Finally, the total flux was calculated following Eq. (17) (Massad et al., 2010). NH<sub>3</sub> flux is calculated in this framework as a difference between the  $\chi_{zo}$  and  $\chi_a$ , scaled by  $R_a$ .

$$F_{NH_3} = \frac{\chi_{z0} - \chi_a}{R_a} \quad , \tag{17}$$

Total exchange flux (F<sub>NH3</sub>) from the dry deposition inferential model gives the direction and magnitude of NH<sub>3</sub> fluxes.

### 3. Results and Discussion

### 3.1 Simulated bidirectional exchange fluxes of NH<sub>3</sub>

Bidirectional fluxes were simulated using the 30-minute NH<sub>3</sub> concentration data set and in situ meteorological data as inputs to the Massad et al. (2010) model, described above. NH<sub>3</sub> concentration, χ<sub>20</sub>, and fluxes have a strong seasonal cycle in RMNP (see Fig. 4). NH<sub>3</sub> flux direction is determined by the difference between χ<sub>20</sub> and χ<sub>a</sub> (Fig. 4a). When NH<sub>3</sub> concentration exceeds the compensation point, NH<sub>3</sub> is deposited to the surface (a negative flux value). Both NH<sub>3</sub> concentrations and deposition fluxes tend to be greatest during the summer (June, July, and August), with 47% of NH<sub>3</sub> modeled annual dry deposition occurring during June, July, and August. NH<sub>3</sub> fluxes also had the largest variability in the summer. Deposition in the spring (March, April, and May) closely follows, with 43% of NH<sub>3</sub> modeled annual dry deposition occurring during March, April, and May. During all seasons, there are periods of net emission from the surface (Fig. 4b). The largest periods of net emission occur in the summer. Daily NH<sub>3</sub> emission fluxes are most common in the winter (December, January, and February), although they are typically smaller than deposition fluxes in the spring and summer.

Figure 4. Daily mean values of: (a.) Daily mean  $\chi_a$  and  $\chi_{z0}$ , and (b.) NH<sub>3</sub> flux.

Total modeled NH<sub>3</sub> flux can be broken down into stomatal, ground, and cuticular fluxes. Figure 5 shows the distribution of simulated NH<sub>3</sub> fluxes for each of these components.

Deposition is driven primarily by stomatal and cuticular fluxes, while ground emission fluxes are sometimes observed. Winter periods of net emission (see Fig. 4b) are driven by the ground flux. One potential limitation of the model used for simulations is that it does not consider snow cover on the ground, which could alter winter fluxes in RMNP.

Figure 5. Total NH<sub>3</sub> simulated fluxes are separated into their component fluxes (stomatal, ground, and cuticular). Simulated fluxes are shown for the entire study period. Boxes show the 25<sup>th</sup> and 75<sup>th</sup> percentiles, and whiskers are determined at 1.5 times the interquartile range.

NH<sub>3</sub> concentrations at RMNP are impacted by emission and transport patterns, which can both increase daytime NH<sub>3</sub> concentrations. NH<sub>3</sub> emissions from agricultural sources have a strong diel pattern driven by volatilization during warmer daytime temperatures. At RMNP, transport from these regions is driven on many days by the mountain-plains circulation, which begins in the late morning and transports polluted air masses westward and upslope to the park (Gebhart et al., 2011). Previous studies have demonstrated that the upslope transport from sources in the Front Range has impacts on deposition and air concentrations in RMNP (Benedict et al., 2018; Pan et al., 2021). During this study, the largest χ<sub>a</sub> values are also observed during upslope transport from source regions in the CO Front Range. These source regions likely disproportionately contribute to NH<sub>3</sub> dry deposition because the difference between  $\chi_a$  and  $\chi_{z0}$  drives the sign and magnitude of the NH<sub>3</sub> flux. On mornings following overnight dew formation, local volatilization from evaporating dew has also been shown to increase morning NH<sub>3</sub> concentrations (Wentworth et al., 2016). This phenomenon was observed in RMNP and corresponds to the increase in the NH<sub>3</sub> diel pattern around 10:00 observed in Fig. 6a. One limitation of the bidirectional flux model used is that NH<sub>3</sub> uptake and emission from dew are not simulated. NH<sub>3</sub> concentration, compensation point, and simulated fluxes each have a strong diel pattern, which peaks during the middle of the day (see Fig. 6). The peak value typically occurs close to 13:00. The soil temperature diel pattern contributes to a higher  $\chi_{z0}$  during the middle of the day. The annual cycle of soil temperature also contributes to the higher  $\chi_{20}$  observed in summer. Although both NH<sub>3</sub> concentration and compensation point peak during midday, we also observe peak deposition fluxes during the middle of the day, indicating that the influence of the diel pattern of NH<sub>3</sub> concentration is stronger than that of the compensation point diel pattern.

Figure 6. Diel pattern of: (a.) NH<sub>3</sub> concentration, (b.) simulated  $\chi_{z0}$ , and (c.) NH<sub>3</sub> fluxes are shown for the full study period in RMNP. Boxes show the 25<sup>th</sup> and 75<sup>th</sup> percentiles, and whiskers are determined at 1.5 times the interquartile range.

To understand the relative importance of NH<sub>3</sub> deposition in RMNP, NH<sub>3</sub> flux simulation results are combined with NADP/NTN wet deposition fluxes and dry deposition fluxes for particulate ammonium (NH<sub>4</sub><sup>+</sup>) and nitrate (NO<sub>3</sub><sup>-</sup>) and gaseous HNO<sub>3</sub> derived from CASTNET concentration observations and MLM deposition velocities, to construct an updated seasonal and annual budget of inorganic N deposition at RMNP. This N<sub>r</sub> deposition budget for all measured inorganic species is shown in Fig. 7a. Due to the lack of current measurements, wet and dry deposition of organic nitrogen are not included. Benedict et al. (2013b) reported annual organic nitrogen wet deposition of 0.6 kg N ha<sup>-1</sup> yr<sup>-1</sup> during their 2008-2009 study. NH<sub>3</sub> dry deposition is the net surface flux from the simulations using 30-minute NH<sub>3</sub> concentration. The inorganic annual N<sub>r</sub> deposition budget totals 3.4 kg N ha<sup>-1</sup> yr<sup>-1</sup>, with the largest contributions coming from NH<sub>4</sub><sup>+</sup> wet deposition (1.34 kg N ha<sup>-1</sup> yr<sup>-1</sup>), NH<sub>3</sub> net dry deposition (0.12 kg N ha<sup>-1</sup> yr<sup>-1</sup>), NO<sub>3</sub><sup>-</sup> wet deposition (0.71 kg N ha<sup>-1</sup> yr<sup>-1</sup>), and HNO<sub>3</sub> dry deposition (0.33 kg N ha<sup>-1</sup> yr<sup>-1</sup>). Overall, reduced N<sub>r</sub> deposition comprises 59% of the total inorganic N deposition to RMNP. NH<sub>3</sub> dry deposition comprises 4% of total inorganic N<sub>r</sub> deposition. Simulated NH<sub>3</sub> dry deposition (0.11 kg N ha<sup>-1</sup> yr<sup>-1</sup>) is smaller than the value estimated by Benedict et al. (2013b) during their 2008-2009 study (0.66 kg N ha<sup>-1</sup> yr<sup>-1</sup>). The previous value estimated NH<sub>3</sub> dry deposition velocity by scaling the HNO<sub>3</sub> dry deposition velocity by 0.7, instead of simulating the bidirectional exchange of NH<sub>3</sub>.

Figure 7. Reactive nitrogen deposition is shown for all species with measured concentrations or deposition for the full year of study. Wet deposition data are from the NADP NTN site at Beaver Meadows. NH<sub>3</sub> dry deposition is modeled using the bidirectional framework from Massad et. al (2010) and 30-minute NH<sub>3</sub> concentration data. Dry deposition of HNO<sub>3</sub> (g), NH<sub>4</sub><sup>+</sup> (p), and NO<sub>3</sub><sup>-</sup> (p) are calculated from the nearby CASTNET site concentration data and deposition velocities from the U.S. EPA MLM. Panel (a.) has the annual deposition of all measured species. Panel (b.) has deposition of all measured N<sub>r</sub> species grouped by month. Only one period of wet deposition was collected by the NADP NTN site during November 2021.

Speciated monthly dry deposition is plotted in Fig. 7b to probe the seasonality of N<sub>r</sub> deposition in RMNP. Net dry deposition of NH<sub>3</sub> was largest during May and August. Total inorganic N<sub>r</sub> deposition peaked during May, due to increased wet deposition. Reduced N<sub>r</sub> deposition exceeded oxidized N<sub>r</sub> deposition in October, December, February, March, April, May, July, and August. Excluding November, where only one period of wet deposition was recorded by the NADP NTN site, reduced N<sub>r</sub> deposition had a fractional contribution ranging from 43 to 74%. In November and January, net NH<sub>3</sub> emission was estimated from the surface.

### 3.2 Impacts of biweekly NH<sub>3</sub> concentration data on simulated fluxes

The use of low time-resolution NH<sub>3</sub> concentrations for flux simulations can produce a low bias compared with fluxes simulated using higher time-resolution NH<sub>3</sub> concentrations. Simulated NH<sub>3</sub> fluxes have a strong diel pattern when simulated at 30-minute resolution (see Fig. 6c), due to changes in NH<sub>3</sub> concentration and meteorology. These complex dynamics are averaged out when an average NH<sub>3</sub> concentration is used, which leads to an underestimation in deposition. Here, we demonstrate that a site-specific correction can be generated to account for the bias introduced by lower time-resolution NH<sub>3</sub> concentration data. Our methods differ from Schrader et al. (2018) in 3 major ways: (i) in situ data is used for both the higher frequency, 30-minute NH<sub>3</sub> concentration, and meteorology, (ii) biweekly passive NH<sub>3</sub> data is used instead of monthly NH<sub>3</sub> data, and (iii) Massad et al. (2010) is used as described instead of using a simplified parameterization. The results of the 30-minute NH<sub>3</sub> and Biweekly NH<sub>3</sub> bidirectional NH<sub>3</sub> flux simulations are compared to generate a site-specific factor to correct for any low bias in the lower

time-resolution flux calculations. Simulated fluxes at biweekly time-resolution (Fig. 8) using the two NH<sub>3</sub> concentration data sets are well correlated ( $R^2 = 0.88$ ), and the NH<sub>3</sub> flux simulation using biweekly integrated NH<sub>3</sub> data can be corrected to match the control flux simulation using a linear fit (slope = 1.03, y-intercept = -1.689). As noted above, RMNP has few two-week periods of net NH<sub>3</sub> emission, and the efficacy of this method should be confirmed at a location with more extensive periods of net NH<sub>3</sub> emission. In particular, NH<sub>3</sub> fluxes above managed agricultural land could differ significantly from the pattern observed in RMNP. This study also focused on fluxes above a forest canopy, and results could differ for grassland ecosystems, which also occur in RMNP. To determine the efficacy in other locations, future investigations should select several sites with different land surface types and NH<sub>3</sub> concentrations to make biweekly and high time-resolution measurements for a year.

Figure 8. Bidirectional NH<sub>3</sub> flux simulated at 30-minute resolution is plotted for 30-minute NH<sub>3</sub> concentration data and biweekly integrated NH<sub>3</sub> concentration data. Fluxes are given as net flux over a two-week period. The least squares linear regression is plotted for the data.

Considering the net flux of NH<sub>3</sub> across the full study period, using the best available time-resolution of 30 minutes, we find a total annual net NH<sub>3</sub> dry deposition flux of 0.11 kg N ha<sup>-1</sup> yr<sup>-1</sup> (Fig. 9). The estimated NH<sub>3</sub> dry deposition drops by 45% to 0.06 kg N ha<sup>-1</sup> yr<sup>-1</sup> using biweekly vs. 30-minute NH<sub>3</sub> concentration measurements. The annual NH<sub>3</sub> dry deposition flux increases to 0.78 kg N ha<sup>-1</sup> yr<sup>-1</sup> when simulating fluxes in a deposition-only unidirectional framework where the NH<sub>3</sub> deposition velocity is scaled as 0.7 times the nitric acid deposition velocity (estimated by the US EPA MLM), an approach previously used for RMNP N deposition budgets (Beem et al., 2010; Benedict et al., 2013a; Benedict et al., 2013b).

Figure 9. Annual NH<sub>3</sub> dry deposition at the NEON Flux Tower in RMNP is shown for three bidirectional simulations using three sets of NH<sub>3</sub> concentration data (30-minute NH<sub>3</sub>, Biweekly NH<sub>3</sub>, and Average Diel Pattern NH<sub>3</sub>) and one unidirectional simulation. Each simulation was run at 30-minute time steps with meteorological parameters from the NEON Flux Tower. The unidirectional NH<sub>3</sub> flux is calculated using biweekly NH<sub>3</sub> concentrations. NH<sub>3</sub> deposition velocities are calculated as 0.7 times the HNO<sub>3</sub> deposition velocity from the U.S. EPA MLM.

Bidirectional flux simulations using biweekly NH<sub>3</sub> data with an average diel pattern of NH<sub>3</sub> yield the same annual NH<sub>3</sub> dry deposition flux as the simulations run using 30-minute NH<sub>3</sub> concentration. This indicates that capturing daily variability in NH<sub>3</sub> concentration profiles is not critical to accurately simulating the annual NH<sub>3</sub> flux. Application of an annual averaged diel pattern misses the highest NH<sub>3</sub> concentrations (Fig. 10); however, across a full year of data, the diel pattern effectively captures the net surface flux. Despite the scatter in Fig. 10a, fluxes simulated with an average diel pattern NH<sub>3</sub> data set are well correlated with simulations using 30-minute NH<sub>3</sub> concentrations (*R*<sup>2</sup> = 0.59) and have a fit close to unity. The daily mean fluxes (Fig. 10b and Fig. 10c) of each simulation have similar seasonal patterns, with periods of net emission and deposition aligned between simulations.

Figure 10. NH<sub>3</sub> fluxes simulated with 30-minute NH<sub>3</sub> concentrations and annual average diel pattern NH<sub>3</sub> concentrations are shown for the full year of data. Panel (a.) directly compares 30-minute simulated fluxes for each data set. Panels (b.) and (c.) show the daily mean fluxes for simulations with 30-minute NH<sub>3</sub> concentration and average diel pattern NH<sub>3</sub> concentration, respectively.

At RMNP, there is a large daily variability in concentration due especially to changes in upslope transport. When an air mass arrives from the Colorado Front Range and NE Colorado, NH<sub>3</sub> concentrations rise significantly due to the large emission sources upwind. For the comparison shown in Fig. 10, the diel pattern was determined using a full year of NH<sub>3</sub> concentration data. Fluxes were also simulated using diel patterns determined with only a month of data, to probe the necessary length of measurements to generate an effective diel pattern. Annual deposition from all flux simulations using each different monthly diel pattern fell within 2% of the annual deposition using the annual average diel pattern. Therefore, in RMNP, one month of 30-minute measurements appears sufficient to generate a diel pattern that will effectively correct the net NH<sub>3</sub> surface flux. Other locations may have larger and/or more complex variability in NH<sub>3</sub> diel patterns and may require longer periods of data collection to establish an NH<sub>3</sub> diel pattern.

# 3.3 Impacts of reanalysis meteorological data on simulated NH<sub>3</sub> fluxes

Bidirectional exchange models require several meteorological and soil parameters, which may not be readily available for many locations of interest. Reanalysis data can provide meteorological inputs for locations where required in situ meteorological and soil measurements are unavailable. To examine the impact on flux simulation accuracy resulting from this substitution at RMNP, the same simulations of NH<sub>3</sub> bidirectional fluxes were run using ERA5 meteorology and soil data. 30-minute NH<sub>3</sub> simulations run with reanalysis data inputs are well correlated ( $R^2 = 0.76$ ) with 30-minute NH<sub>3</sub> simulations run with in situ data inputs (see Fig. 11), but overestimate the annual NH<sub>3</sub> deposition flux by a factor of 2. From Fig. 11, we find that the use of ERA5 reanalysis data in the simulation of NH<sub>3</sub> bidirectional fluxes introduces a low bias to the flux magnitude in RMNP compared to in situ meteorological data, for both positive (emission) and negative (deposition) fluxes. However,

because the decrease in deposition fluxes is smaller than the decrease in emission fluxes, we observe an annual overestimation from simulations using ERA5.

Figure 11. Bidirectional NH<sub>3</sub> flux simulated with ERA5 meteorology and NEON meteorology at 30-minute resolution using the 30-minute NH<sub>3</sub> concentration. The least squares linear regression is plotted for the data in red.

The low bias for fluxes simulated using ERA5 reanalysis data is investigated further to explore what parameters influence this bias. Net NH<sub>3</sub> fluxes are simulated using Eq. (17), which relies on  $\chi_{z0}$ , NH<sub>3</sub> concentration, and R<sub>a</sub>. We find that the simulations using reanalysis data generate  $\chi_{z0}$ , which agree well with the simulations using in situ measurements (*slope* = 1.03,  $R^2$  = 0.98).

Figure 12. Aerodynamic resistances are shown for simulations using in situ meteorological data from the NEON flux tower and reanalysis meteorological data from ERA5. The diel patterns are shown in panels (a.) and (b.), respectively. Panel (c.) directly compares simulated  $R_a$  values using NEON in situ and ERA5 reanalysis data.

Although the general diel pattern of R<sub>a</sub> is well captured using reanalysis data, R<sub>a</sub> magnitudes differ substantially between the two simulations (Fig. 12a and 12b), with the largest difference occurring overnight. Maximum R<sub>a</sub> values from the reanalysis simulations are an order of magnitude larger than those derived using in situ meteorology. A comparison of the two data sets shows (Fig. 12c) a typical enhancement of approximately a factor of four. Increased R<sub>a</sub> values result in lower simulated NH<sub>3</sub>

fluxes. The R<sub>a</sub> bias is likely driven by differences in the u\* and L, which are used to calculate R<sub>a</sub>. ERA5 data underestimates u\* by a factor of 5 when compared with the in situ NEON data (*slope* = 0.2). The in situ NEON data also sets a minimum u\* value (0.2 m s<sup>-1</sup>), while the ERA5 data allows u\* values below 0.2 m s<sup>-1</sup>. Comparisons of all meteorological parameters used can be found in the supplementary information. This discrepancy in modeled R<sub>a</sub> may be due to the gridded nature of reanalysis data, which represents a large area of variable land types and complex topography using only a single value (Hogrefe et al., 2023). Previous work has identified heat and moisture fluxes as large areas of uncertainty in ERA5 Reanalysis (Kong et al., 2022; Mayer et al., 2022). Two case studies were conducted to probe the relative importance of u\* and L. The case studies are described in the supplementary information. Differences in R<sub>a</sub> were impacted by both u\* and L, accounting for 23% and 10%, respectively, of the discrepancy between in situ and ERA5 flux simulations.

# 4. Conclusion

Fluxes of NH<sub>3</sub> (g) can be simulated using a bidirectional model, which uses rapidly changing meteorology paired with air concentrations and soil parameters to infer flux direction and magnitude. We use a bidirectional NH<sub>3</sub> flux model, proposed by Massad et al. (2010), to simulate a year of NH<sub>3</sub> fluxes above a subalpine forest ecosystem in Rocky Mountain National Park. The net NH<sub>3</sub> dry deposition to the ecosystem is estimated at 0.11 kg N ha<sup>-1</sup> yr<sup>-1</sup>, comprising 4% of total inorganic reactive nitrogen deposition. This is significantly lower than previous estimates for RMNP, which did not consider the bidirectional nature of the exchange. Due to the observed low bias in passive NH<sub>3</sub> observations and the sensitivity of simulations to NH<sub>3</sub> concentrations, this is likely a low bound. The sensitivity of NH<sub>3</sub> flux modelling to  $\chi_a$  was tested by scaling the input concentration by 9% to account for the error discussed in Puchalski et al. (2011). This resulted in an annual deposition increase of 47%, indicating the importance of accurate NH<sub>3</sub> measurements for flux modelling. Additionally, since the highest NH<sub>3</sub> concentrations occur during upslope events, the sources contributing to these events likely have a disproportionate effect on deposition. One limitation of this model is the exclusion of snow cover, which could significantly change NH<sub>3</sub> fluxes in the winter, when RMNP has frequent snow events. To probe the impact of snow cover, a sensitivity test was conducted setting  $\chi_g$  equal to zero during the winter (December, January, and February), which increased annual deposition by 0.06 kg N ha<sup>-1</sup> yr<sup>-1</sup>. However, this analysis does not take into account how the surface differences may change NH<sub>3</sub> fluxes above snow. Future work should investigate NH<sub>3</sub> fluxes above snow cover to better simulate the exchange of NH<sub>3</sub> in regions with snow.

Due to the cost and technical challenges of making continual, high time-resolution NH<sub>3</sub> concentration measurements, there is growing interest in using integrated biweekly passive NH<sub>3</sub> measurements, such as those from the NADP AMoN network, for flux simulations. Here, we establish that a site-specific correction can be used to correct a bias introduced by using lower time-resolution passive NH<sub>3</sub> measurements over the studied forest canopy in RMNP. We also establish that an average NH<sub>3</sub> diel pattern can be used to interpolate 30-minute NH<sub>3</sub> concentration and correct for the bias introduced by passive NH<sub>3</sub> measurements. In RMNP, a month of measurements proved sufficient to determine the diel pattern used for flux simulations. The correction factor and diel pattern, however, likely vary by location due to differences in ecosystem characteristics and

factors influencing NH<sub>3</sub> concentrations. Due to the potential regional differences and changes associated with land surface type, additional sites should be studied to assess the impact of measurement time-resolution on NH<sub>3</sub> flux simulations. To understand the seasonal variability in diel pattern and efficacy of diel pattern application for flux simulations, measurements should be conducted for a full year.

Local micrometeorological and soil measurements are also frequently unavailable, making the use of reanalysis data a desirable alternative for NH<sub>3</sub> flux simulations. In our location, the use of reanalysis data adds a bias that leads to overestimates of net NH<sub>3</sub> deposition. We found it was possible to apply a correction to address this bias, but this factor likely varies by location, in particular over different land surface types within a reanalysis grid cell. Future studies should explore the relationship between in situ measurements and reanalysis products above different land surface types, varied topography, and in different regions. Understanding how to correct for the biases introduced through the use of reanalysis data would allow improved modelling of NH<sub>3</sub> bidirectional fluxes in regions lacking high time-resolution measurements.

In this analysis, we simulated the bidirectional exchange of NH<sub>3</sub> above a forest ecosystem using the model proposed in Massad et al. (2010). However, there are other bidirectional exchange models (e.g., Zhang et al., 2010; Pleim et al., 2013) and their simulated fluxes may differ significantly from the model used here (Jongenelen et al., 2025). In the bidirectional exchange model used here, we observe that the selected inputs for NH<sub>3</sub> concentration and meteorological data may introduce biases into the simulated NH<sub>3</sub> fluxes. This may also be true for the other models when simulating NH<sub>3</sub> bidirectional exchange, a good topic for future research.

# 515 Data Availability

The ammonia concentration data used in the study is available at DOI: 10.5061/dryad.0cfxpnwcw. The NEON flux tower eddy covariance data bundle is available at: <a href="https://data.neonscience.org/data-products/DP4.00200.001">https://data.neonscience.org/data-products/DP4.00200.001</a>. ERA5 reanalysis data is available at: <a href="https://www.ecmwf.int/en/forecasts/dataset/ecmwf-reanalysis-v5">https://www.ecmwf.int/en/forecasts/dataset/ecmwf-reanalysis-v5</a>. CASTNET data are available at: <a href="https://madp.slh.wisc.edu/networks/national-trends-network/">https://madp.slh.wisc.edu/networks/national-trends-network/</a>.

### **Author Contributions**

JC, BS, DP, and JW designed the measurement campaign. LN, AS, and DP made and processed Rocky Mountain National Park measurements. DP developed the model code. LN designed and ran the bidirectional exchange simulations. LN prepared the manuscript with contributions from all co-authors.

# 525 Competing interest

The authors declare that they have no conflict of interest.

### Disclaimer

The results contain modified Copernicus Climate Change Service information 2020. Neither the European Commission nor ECMWF is responsible for any use that may be made of the Copernicus information or data it contains. The results make use of data collected by the CASTNET program from the U.S. Environmental Protection Agency. The views expressed are of the authors and do not necessarily reflect those of the U.S. EPA or any other organizations that the data used was obtained from.

### Acknowledgments

The authors would also like to thank the NEON team for their support in collecting biweekly passive NH<sub>3</sub> data. The authors thank Nikolas Tafoya for his assistance in collecting measurements on the NEON Tower.

### 535 Financial support

This work was supported by the U.S. Environmental Protection Agency Regional Applied Research Efforts Program, Project #2237, and the National Park Service (NPS) under Agreement Number P20AC00679 with Colorado State University.

. This material is based in part upon work supported by the National Science Foundation through the National Ecological Observatory Network Program. The NEON Program is operated under a cooperative agreement by Battelle.

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
