# Peer review of "Sensitivity of Simulated Ammonia Fluxes in Rocky Mountain National Park to Measurement Time Resolution and Meteorological Inputs"

_EGUsphere, 2025_

## Author Comment (AC1)

**We appreciate the helpful comments from the reviewers and the editor that have helped to improve the quality of our revised manuscript. Our response to reviewer and editor comments is included below in RED**

**RC1:** ['Comment on egusphere-2025-1167'](), **Anonymous Referee #1, 03 Apr 2025**

General comments:

The authors present a modelling study that investigates the impact of using low-resolution concentration data for computing the ammonia dry deposition flux in a forest ecosystem in the Rocky Mountains National Park. Among the results, a key finding is that using the low-resolution ammonia concentration data led to an underestimation of the dry deposition flux. Additionally, a correction factor is derived, which can be used to mitigate this bias. The case study could provide an interesting continuation of the work by Schrader et al. (2018) but omits relevant methodological details and additionally requires extra proofreading.

We appreciate the reviewer's assessment and respond to individual comments below.

Specific comments:

Paragraph at lines 18 – 19: The wording "[…] from more commonly available input data was evaluated" is unclear. It would be clearer to directly state which data have been used instead – in this case, the bi-weekly ammonia measurements and the ERA5 meteorological data.

The text was updated to specify that we evaluated the impact of biweekly $NH_3$ measurements and ERA5 meteorology.

Line 27: Perhaps "$NH_x$ ($NH_3 + NH_4^+$) emissions" instead of "$NH_3$ emissions".

Primary emissions of $NH_x$ are gas-phase $NH_3$. Particle ammonium is formed through phase partitioning involving salt formation with acids in the atmosphere. Here, we are specifically talking about the direct emissions that lead to reactive nitrogen deposition.

Line 33: Eutrophication is not limited to lakes only, so consider omitting the word 'lake'.

Good point. Lake effects have been one of the major indicators of excess N deposition impacts for RMNP. However, to make it more generally applicable, we have removed the word "lake" to include all eutrophication effects.

Figure 1: Given the relevant mountain-plains circulation taking place in the Rocky Mountain National Park and its relevance to the $NH_3$ concentrations, this figure would benefit from an elevation map. Additionally, a scale should be inserted.

Figure 1 has been updated to include an elevation profile and a scale bar.

Section 2.2.1: The leaf-area-index (LAI) is an important variable in the modeling of $NH_3$ atmosphere-biosphere exchange and should also be mentioned in this section.

LAI was estimated using the NEON LAI spectrometer mosaic product, which is derived from remote sensing data. The main text has been updated to indicate where the LAI came from. An

additional figure has been added to the SI to illustrate the LAI selection and show the spatial variability of LAI values in this area. A sensitivity analysis has also been added to the SI to give the reader some insight into the effects on $NH_3$ fluxes from changing LAI.

Section 2.2.1: This section would improve by shortly characterizing the typical meteorological conditions at the NEON site (e.g., average temperature, relative humidity, amount of rain days, etc.) as well as the average $NH_3$ concentration. Moreover, the number of days with snow might be relevant here, as $NH_3$ exchange differs when there is snow present.

Due to its high elevation location, the meteorology at the NEON tower in RMNP is highly variable. Additional context has been added to this section to give the reader a general sense of the typical meteorology in RMNP.

Lines 112 – 114: What is the name of the instrument measuring the friction velocity, and what is the temporal resolution of this instrument?

NEON calculates friction velocity (u*) using 3D sonic anemometers, which have a resolution of 20 Hz.

Section 2.3.1: Please mention the number of bi-weekly $NH_3$ concentration measurements that have been collected.

The number of biweekly sampling periods (27) is now included in the text.

Section 2.3.2: I am currently missing information on the quality control of the measurements. For example, when the atmosphere is stable, stratification of the atmosphere occurs, which can hinder accurate $NH_3$ concentration measurements as the atmosphere is not well-mixed. Additionally, similarly to the comment for Section 2.3.1, please mention the number of half-hourly $NH_3$ measurements made with the AirSentry and provide information on how many observations have been filtered, if any.

For $NH_3$ concentration, we could see lower concentrations at the surface due to stratification of the atmosphere. However, the exchange model used only considers concentration at the reference height. During our campaign, we also measured $NH_3$ gradients on the NEON tower which will be used in a later work to directly determine $NH_3$ fluxes. The observed gradients may give insight into the effects when $NH_3$ is not well mixed in the atmosphere. Elevated $NH_3$ concentrations are brought to RMNP by upslope transport, where the winds would contribute to atmospheric mixing. For quality control of $NH_3$ measurements, the number of AirSentry and passive measurements is now included. Only $NH_3$ data missing due to power outages have been removed from the AirSentry dataset.

Lines 145 – 147: Please provide the full form of the abbreviation 'NPS' when it is first mentioned.

The abbreviation "NPS" for National Park Service is now defined in the text, where first mentioned.

Section 2.3.3: I think that this section can benefit from a table summarizing the specifications of each of the three datasets (e.g., location of measurement, sampling type, sample size, temporal resolution, nomenclature) to both summarize the three different datasets and to guide the reader through the differences between the datasets.

We are concerned that some of the information listed may be confusing to the reader. For example, including "location of measurement" for the 30-minute data product implies that the raw $NH_3$ concentration was taken from the AirSentry location. A bulleted list in this section summarizes the data products and nomenclature. For the key modeling understanding, it is most important that the reader understands the respective time resolution and therefore impacts of time resolution on model results.

Figure 2: The caption of Figure 2 repeats text from the main body and could, therefore be omitted.

The caption of Figure 2 has been edited to remove repeated information. The definitions of each dataset are now included in the main text only.

Lines 170 – 171: Have you considered that the diurnal $NH_3$ concentration cycle at the NEON site could be different compared to the diurnal cycle at the NPS shelter, related to differences such as the physical location of the measurements and the vegetation type at the two sites? Regarding the latter, deposition velocities can be lower above grasslands compared to forests, given the lower roughness length $z_0$ of grasslands, which can consequently lead to higher $NH_3$ concentrations above grasslands. This section or the discussion should at least contain a more critical evaluation concerning the systematic differences in the diurnal cycle above grasslands and forest sites.

Yes, we did consider how the diel pattern of $NH_3$ could vary between the grassland site and NEON site. From August 23, 2021, to October 4, 2021, we deployed University Research Glassware annual denuders to measure $NH_3$ on the NEON tower. These data are compared to the raw AirSentry data and the AirSentry data scaled to the passive measurements in Fig S1. We find that the daytime $NH_3$ concentrations agree well between the sites. Overnight URG samples generally have higher concentration than what we observe in the AirSentry data. Additional discussion has been added to the main text to explain the potential differences.

Lines 230 – 232: For the sake of completeness, it may be helpful to include at least Eq. (15) from Massad et al. (2010).

Eq. (15) has been added to the text. The wording of the section has also been improved to clarify the modelled resistances.

Additionally, the canopy height at which the wind speed is measured is 11 m, while the mean canopy height mentioned in Section 2.2.1 is 19 m. This difference should be addressed to avoid confusion.

Thank you for catching this! The wind speed was taken from the top of the tower, not the canopy height of 19 m. The noted height of 11 m was a typo. The text has been updated to reflect the proper height of wind speed measurements.

Lines 239 – 243: Massad et al. (2010) provide corrections for the temperature and leaf-area index when calculating the cuticular resistance $R_w$, based on the findings by Flechard et al. (2010) and Zhang et al. (2003). For example, Schrader et al. (2016) also incorporate these effects in the $R_w$ parameterization as shown in Eq. (5) in their paper. If you choose to omit the LAI and temperature coefficient from the $R_w$ parameterization, that decision should be justified.

There are two subsections in Massad et al. (2010) that discuss Rw. In Section 2.2, Equations (3 – 4) include the T and LAI effects in Rw(corr). In Section 4.6, they proposed a generalized formulation for Rw(corr) (Eq. (24)), where the T and LAI effects are removed. We followed this generalized formulation, which does not include T and LAI. However, upon reviewing Massad et al. (2010) again, we found that Table 8 contradicted Equation (24). It appears that Rw(corr) from Equation (24) might be Rw, meaning it would still require T and LAI corrections. Since we do not have the original data used for the generalization, we cannot investigate which form is correct, and correcting the Rw parameterization is beyond our scope.

Line 250-251: Given the importance of the stomatal emission potential, it would be appropriate to introduce what emission potentials are. Moreover, please discuss how and from which equation the emission potential of 4 has been derived. This value does seem rather low.

The text has been updated to include a brief conceptual introduction for emission potential. The value of 4 for stomatal emission potential was used initially to match measurements from other regions with very low annual ecosystem N input. We have updated this value based on foliage measurements taken around the NEON site in RMNP. The updated stomatal emission potential is a weighted average of the species-specific emission potential and average land coverage at the site. An explanation of the sample collection is provided in the supplementary information. The stomatal emission potential now used for all simulations is 29.

Line 253 – 259: The parameterization by Massad et al. (2010) does not originally calculate the soil compensation point or the soil resistance ($r_g$ + $r_{ac}$). Often, the exchange of gases with the soil is not taken into account in dry deposition schemes due to the overlying canopy, which will (re)capture $NH_3$. Moreover, Massad et al. (2010) state that very few data is available regarding ground layer emissions. Thus, please elaborate why $NH_3$ exchange with the soil is modeled here.

In Massad et al. (2010), $\chi_g$ is called the ground compensation point (section 2.4) and is included in the Eq. (12) calculation of $\chi_c$. The text has been updated to make the nomenclature consistent with that described in Massad et al. (2010) to avoid confusion. $R_g$, $R_{ac}$ and $\chi_g$ can be found in Fig. 1, schematic in Massad et al. (2010). You raise a good point that soil emissions could be recaptured by the canopy above. In Massad et al (2010), they suggest using a soil emission potential of zero in unmanaged ecosystems. However, this has more to do with the lack of soil measurements. In RMNP, we are fortunate to have measurements to base our soil emission potential on. Due to this, we have included the effects of soil in our analysis.

Line 262: $z_0$ is not the reference height but the roughness length. This mislabeling occurs more often, both in text and in figures.

The text has been updated to be consistent throughout and properly labels the reference height (z) and roughness length (z0).

Line 265: $\chi_a$ is mentioned here for the first time, so it requires a brief explanation.

An explanation of $\chi_a$ is now included in the revised manuscript.

Line 269: Connected to the comment at line 265, here, the ammonia concentration is denoted as $[NH_3]$ instead of $\chi_a$. For the sake of consistency, use a single notation for atmospheric ammonia throughout the manuscript.

The text and figures have been updated to use consistent notation for atmospheric ammonia concentration. In all locations, the atmospheric ammonia concentration is denoted as $\chi_a$.

Moreover, the denominator contains the term "$\cdot\ 10^3$" which is not included in the original parameterization by Massad et al. (2010). Please specify why this term is included.

The term "$10^3$" was erroneously included in the text based on a necessary unit conversion for the model simulation results. It has been removed from the text to be consistent with Massad et al. (2010).

Lines 300-303: Can you be certain that the morning increase in $NH_3$ concentration at the site is mainly due to $NH_3$ evaporation from cuticular dew layers, and not also influenced by either the diel mountain-plains circulation transporting polluted air with $NH_3$ or $NH_3$ emission from the stomata?

Previous work from Wentworth et al. (2016) found that the timing of the early morning $NH_3$ emission pulse was temporally correlated with dew evaporation, not transport. The diel pattern of the mountain-plains circulation is typically later in the day than the observed dew emission. Additionally, the early morning $NH_3$ emission pulse was not observed on mornings without dew or during precipitation.

Section 3.2: I am confused here to what extent the same method of Schrader et al. (2018) is applied here. The method by Schrader et al. (2018) proposes a true average $NH_3$ flux formula (Eq. 9 in Schrader et al., 2018) when long-term average $NH_3$ concentrations have been used as input in a dry deposition scheme. Additionally, they provide a method to calculate this true flux, which requires the covariance between the exchange velocity $v_{ex}$ and the atmospheric concentration $\chi_a$ to be calculated. If I understood your method correctly, you have run the dry deposition scheme with the 30-minute concentration data and the bi-weekly sample data and afterwards compared the slope, intercept, and the $R^2$ of the two different flux outputs – which is ultimately used to correct the fluxes. While both your methodology and that of Schrader et al. (2018) aim to correct $NH_3$ flux calculations based on low-temporal-resolution $NH_3$ data, the approaches themselves differ substantially. I recommend rephrasing this, as it currently gives the impression that you applied the exact same methodology, aside from the three exceptions noted in lines 344 – 346.

Thank you for pointing this out. You understood correctly; we aim to correct $NH_3$ fluxes from low-temporal resolution $NH_3$ data as did Schrader et al. (2018). However, we use the Massad et

al. (2010) model applied at high time resolution, instead of an average $NH_3$ flux formula proposed by Schrader et al. (2018). The text has been updated to better reflect the similarities and differences between the two methods.

Finally, the average 30-minute concentrations from the AirSentry have been scaled to match the bi-weekly passive $NH_3$ concentration. There is a high chance that this will improve the $R^2$ and also affect the slope and intercept used for correcting the fluxes. Have you considered the effect this has on the efficacy of your method?

Yes, we did consider the additional impact that could be observed if the mean biweekly concentration was different between sampling techniques and locations. In particular, the observed difference in $NH_3$ concentrations above grassland and forest sites leads us to normalize the values to match what was observed above the forest. For this project, we wanted to specifically understand the impact of changing the time resolution and therefore decided to remove the additional complication of sampling technique variation in concentration.

Lines 363 – 366: Do you have an explanation for why the total $NH_3$ deposition is significantly lower using the 30-minute $NH_3$ concentration data compared to the unidirectional framework? Is this only caused by the inclusion of compensation points or, for example, by differences between the $NH_3$ and $HNO_3$ concentrations at the RMNP?

The total $NH_3$ deposition is significantly lower using the 30-minute $NH_3$ concentration data and bidirectional model compared to the unidirectional framework because of the inclusion of compensation points. For the unidirectional deposition velocities, we are using a fraction of the modeled $HNO_3$ deposition velocity, so the relative concentration of $NH_3$ and $HNO_3$ would not affect the relative deposition.

Line 395 – 397: "[…] but overestimate the annual $NH_3$ deposition flux by 59%". Please indicate which $NH_3$ deposition calculation is used as a reference here (i.e., either the $HNO_3$-based calculation of the unidirectional model or the total $NH_3$ deposition based on the 30-minute $NH_3$ concentration data).

The text now reads: "30-minute $NH_3$ simulations run with reanalysis data inputs are well correlated ($R^2$ = 0.77) with 30-minute $NH_3$ simulations run with in situ data inputs (see Fig. 11) but overestimate the annual $NH_3$ deposition flux", to indicate that the difference is based on the 30-minute $NH_3$ data and bidirectional simulations.

Additionally, while line 397 states an "overestimation" of the $NH_3$ flux when using the ERA5 meteorological data, I think this is supposed to be an underestimation of the $NH_3$ flux, as the deposition strength decreases caused by the higher $R_a$.

This is an important distinction, which we have made more clear in the text. Between the two meteorological input simulations, $R_a$ differences reduce the magnitude of ERA5 simulations. However, when we consider the annual net effect, the change to negative fluxes is smaller than the change to positive fluxes. Therefore, the annual $NH_3$ dry deposition is overestimated by ERA5.

Technical corrections:

All headers: Titles should only contain capitalization for the first word and proper nouns.

All headers have been updated to remove erroneous capitalization.

Line 105 – 108: Ammonia should be written with a "3" in subscript (i.e., $NH_3$ instead of NH3).

The abbreviation $NH_3$ for ammonia has been updated to include the "3" in subscript for these lines.

Line 250: Replace "equation 10" with Eq. (10)

"Equation 10" has been replaced with Eq. (10).

Line 262: Replace "equation 15" with Eq. (15)

"Equation 15" has been replaced with Eq. (15).

Line 266: Replace "equation 16" with Eq. (16)

"Equation 16" has been replaced with Eq. (16).

Line 297: Fig. 11 does not have a subfigure 'a'.

Thank you. In the previous version, this figure contained 2 subplots. The 'a' has been removed when referencing Fig. 11 for clarity.

Lines 415-417: The phrase "Maximum $R_a$ values from the reanalysis simulations are greater than an order of magnitude larger […]" could benefit from improved sentence structure.

That sentence has been reworded and divided into two sentences to make it clearer.

References:

Flechard, C. R., Spirig, C., Neftel, A., and Ammann, C.: The annual ammonia budget of fertilised cut grassland - Part 2: Seasonal variations and compensation point modeling, Biogeosciences, 7, 537–556, https://doi.org/10.5194/bg-7-537-2010, 2010.

Massad, R. S., Nemitz, E., and Sutton, M. A.: Review and parameterisation of bi-directional ammonia exchange between vegetation and the atmosphere, Atmospheric Chem. Phys., 10, 10359–10386, https://doi.org/10.5194/acp-10-10359-2010, 2010.

Schrader, F., Brümmer, C., Flechard, C. R., Kruit, R. J. W., Van Zanten, M. C., Zöll, U., Hensen, A., and Erisman, J. W.: Non-stomatal exchange in ammonia dry deposition models: Comparison of two state-of-the-art approaches, Atmospheric Chem. Phys., 16, 13417–13430, https://doi.org/10.5194/acp-16-13417-2016, 2016.

Zhang, L., Brook, J. R., and Vet, R.: A revised parameterization for gaseous dry deposition in air-quality models, Atmospheric Chem. Phys., 3, 2067–2082, https://doi.org/10.5194/acp-3-2067-2003, 2003.

**RC2**: 'Comment on egusphere-2025-1167', Anonymous Referee #2, 26 Apr 2025

While filling 5.8% of missing data using average diel patterns is pragmatic, this approach assumes temporal homogeneity in $NH_3$ behavior. The authors should quantify the potential error introduced by this method, especially during episodic events (e.g., wildfire plumes or synoptic transport), which may not follow average patterns.

We repeated the bidirectional flux simulations using the maximum and minimum diel pattern to fill the data and found that across the full year of data, it impacted the annual deposition by less than 5%. This indicates that the error introduced by an average diel pattern is relatively small when considering the annual deposition. This is in part due to the correction factor we apply to ensure that the 2-week mean concentration matches that recorded in the passive measurements. As stated here, this could still miss episodic events. However, events with the potential to profoundly impact the annual deposition would be captured in the passive measurements.

The use of Radiello passive samplers, which have a documented low bias, raises questions about the accuracy of biweekly $NH_3$ concentrations. Scaling high-resolution AirSentry data to match passive sampler averages may obscure short-term variability critical for flux simulations. A sensitivity analysis on the scaling method's impact would strengthen confidence.

We scaled the AirSentry data to make sure that the differences were from the effects of measurement time resolution, without mean $NH_3$ changing the comparison. Additionally, there are likely some concentration differences above forest and grassland ecosystems where these two measurements were taken. Puchalski et al. (2011) found a low bias with MRSE of 9% when comparing Radiello passive samplers to other sampling techniques. This suggests that our calculated annual $NH_3$ dry deposition is a lower bound. A sensitivity analysis of $NH_3$ concentration on $NH_3$ fluxes is now included to give readers a sense of the changes associated with reasonable $NH_3$ concentration. Although the passive low bias is only 9%, we find that simulations with the mean value increased by 9% result in an annual deposition that is 47% larger than previously estimated. This illustrates how sensitive $NH_3$ flux simulations are to concentration inputs since the relevant driver is the difference between ambient concentration and compensation point. This additional discussion has been added to the supplementary information. For this paper, we are focused on the specific impacts of time resolution and place it in the context of N deposition in RMNP. We have added more discussion about the impacts of $NH_3$ concentration and specifically discussion of the passive measurement low bias to the manuscript. In addition to this, an upcoming paper will improve our understanding of true annual deposition in RMNP, by using a gradient method to compare with flux results and update model parameterization for this ecosystem.

The exclusion of snow cover effects on surface exchange is a significant oversight, particularly for winter fluxes where snow alters surface-atmosphere interactions. This omission may explain discrepancies in winter emission estimates.

For areas that have snow cover, the omission of snow cover as a parameter for flux simulations is a large limitation. The effects of snow are poorly understood for $NH_3$ fluxes and are not included in the bidirectional model used in this work. The generation of an equation that would capture these effects, to include in the model, is outside of our capabilities using this dataset. Future works should investigate these impacts and directly measure fluxes above snow cover. It could be especially important for ecosystem impacts in regions that experience heavy snowmelt. We have added discussion of the potential impacts of snow cover and potential biases introduced to the conclusions section.

The soil compensation point ($\chi_9$) relies on estimated total ammoniacal nitrogen (TAN = 9.6 mg kg$^{-1}$). No justification or sensitivity analysis for this value is provided, yet it directly influences $\chi_9$ and flux calculations.

The citation was inadvertently left out of the original manuscript. The text has been updated to include a citation to Stratton et al. (2018), who conducted soil analysis in RMNP and reported measurements of soil nitrogen and specifically ammoniacal nitrogen. We have also included a sensitivity test for TAN value in the manuscript and supplementary information.

While a one-month dataset sufficed to derive a diel correction factor in RMNP, this may not hold for regions with stronger seasonal variability (e.g., monsoon-influenced areas). The authors should acknowledge this limitation and recommend longer sampling periods for less-studied ecosystems.

It was not our intent to indicate that this length of sampling would necessarily be effective for all regions, although it tells us something interesting about the seasonality of the diel pattern of $NH_3$ in RMNP. We have added the sentence "Other locations may have larger and/or more complex variability in $NH_3$ diel pattern and may require longer periods of data collection to establish an effective $NH_3$ diel pattern." To address the limitation of only having data from one site. We also added additional description to the conclusion encouraging future studies to collect data for a full year to establish effective diel patterns.

The 31-km resolution of ERA5 likely smooths local topographic effects, critical in mountainous regions like RMNP. While the overestimation of deposition is noted, the paper lacks a quantitative assessment of how terrain complexity biases reanalysis inputs (e.g., friction velocity, Obukhov length).

To complement our analysis of aerodynamic resistance from ERA5 and NEON meteorology, we ran two case studies to directly look at the impact of friction velocity and Obukhov length. This was done by repeating the $NH_3$ flux simulations using the ERA5 meteorology but replacing friction velocity with the value from NEON. This was repeated for Obukhov length. These case studies are now included in the supplementary information. We see the largest impact from our simulation using the NEON Obukhov Length, however neither simulation entirely corrects for the observed differences.

The bidirectional model's annual $NH_3$ deposition (0.17 kg N ha$^{-1}$ yr$^{-1}$) is 74% lower than earlier unidirectional estimates (0.66 kg N ha$^{-1}$ yr$^{-1}$; Benedict et al., 2013b). However, the paper does

not reconcile this stark difference with field measurements or independent validation (e.g., eddy covariance data).

The difference between the previous unidirectional estimate and our bidirectional model estimate is large. For this work, we focused on the impacts of measurement resolution to probe the impacts of time resolution and reanalysis meteorology. We will have another paper published shortly which looks at fluxes simulated using the gradient method in RMNP. Additionally, these gradient fluxes will be compared with bidirectional model simulations.

Critical Load Implications: The 6% $NH_3$ contribution to total N deposition is framed as minor, but RMNP's critical load (1.5 kg N ha$^{-1}$ yr$^{-1}$) is still exceeded by current deposition (3.4 kg N ha$^{-1}$ yr$^{-1}$). The policy relevance of these findings—particularly for targeting emission reductions—deserves deeper discussion.

Critical Loads were developed using only wet deposition of N species. While they are helpful for putting $NH_3$ dry deposition into a greater context in RMNP, it is challenging to place dry deposition into a policy context given the exclusion of dry deposition in Critical Loads. Notably, the source regions that are important for wet deposition are the same as those important for dry deposition. The policy implications are discussed in the context of where elevated concentrations are transported from and the importance of source regions in the CO Front Range to the east of RMNP. Notably, the highest $NH_3$ concentrations are observed during upslope transport from source regions in the CO Front Range. These source regions disproportionately contribute to $NH_3$ dry deposition because the difference between atmospheric concentration and compensation points drives the sign and magnitude of the $NH_3$ flux.

The study focuses on a subalpine forest, but bidirectional flux behavior may differ in grasslands or agricultural areas. The conclusion's recommendation for multi-site validation is appropriate but underdeveloped.

We have added an additional discussion of suggested multi-site validation to the conclusions to better outline how future researchers could employ these techniques to improve bidirectional modeling of $NH_3$ fluxes.

The linear correction for biweekly data (slope = 1.07, $R^2$ = 0.89) works well in RMNP but may fail in regions with frequent emission-dominated periods. A discussion of how site-specific factors (e.g., land use, climate) affect correction efficacy would enhance practical utility.

We have updated the discussion in the section considering the site specific correction factor to:

"As noted above, RMNP has few two-week periods of net $NH_3$ emission, and the efficacy of this method should be confirmed at a location with more extensive periods of net $NH_3$ emission. In particular, $NH_3$ fluxes above managed agricultural land could differ significantly from the pattern observed in RMNP. This study also focused on fluxes above a forest canopy, and results could differ for grassland ecosystems, which also occur in RMNP. To determine the efficacy in other locations, future investigations should select several sites with different land surface types and $NH_3$ concentrations to make biweekly and high-time resolution measurements for a year"

Key figures (e.g., Figure 7) lack clarity in distinguishing reduced vs. oxidized N species in grayscale. Colorblind-friendly palettes or pattern fills would improve readability.

Figure 7 has been updated to a more colorblind-friendly palette and hatching has been added to make the columns distinguishable even in grayscale. Thank you for catching this! We have checked the rest of the figures again to ensure they are colorblind-friendly.

Sections on resistance parameterizations (e.g., Equations 3–8) are dense and could benefit from schematic summaries or appendices to aid non-specialist readers.

The schematic in Fig 3. now includes all of the resistances and compensation points. We have also updated the text to encourage readers to reference Fig. 3 while they are looking at the more dense equations. We hope this will assist readers comprehension of the equations used, which can be quite dense.

Include sensitivity analyses for key parameters (TAN, snow cover, passive sampler scaling).

We have added a sensitivity analysis for key parameters to the supplementary information. This includes $NH_3$ concentration, TAN, and LAI values. We are not able to probe the sensitivity of snow cover because of its lack of inclusion in the model used for simulations.

Validate model outputs against independent flux measurements or isotopic tracers.

For this dataset, we lack measurements of isotopic tracers and long periods of flux measurements. In a future work, we will look at fluxes using a gradient method and compare them with those simulated using bidirectional models.

Expand the discussion on policy implications, particularly for RMNP's nitrogen management.

For this work, we are focused on investigating the sensitivity of simulated fluxes to concentration and meteorology inputs. We have revised the text to further elaborate on how the bidirectional framework enhances the impact of source regions in the Colorado Front Range, as the highest $NH_3$ concentrations are transported from these areas. Additionally, we have added some sensitivity analysis which indicates that our annual $NH_3$ dry deposition is likely a low bound and may need to be updated for RMNP nitrogen management.

Clarify figures and technical sections to improve accessibility for interdisciplinary audiences.

The figures have been updated to verify accessibility. Notably, the colors in Fig 7. have been changed and hatching has been added to improve readability.

This paper makes a meaningful contribution to atmospheric deposition science but requires addressing methodological uncertainties and broadening the discussion to enhance its impact.

We have addressed the methodological uncertainties raised here and included additional sections where relevant to improve the clarity and impact.

**EC1**: 'Comment on egusphere-2025-1167', Leiming Zhang, 28 May 2025

I have the following comments for you to consider when revising your manuscript:

A recent study by Jongenelen et al. (https://doi.org/10.5194/acp-25-4943-2025) demonstrated very large uncertainties in the modeled ammonia flux between using three existing bi-directional exchange models, one of which is chosen in your study. Can the major findings presented in your study be generalized if a different bi-directional flux exchange model is used?

Our findings about the impact of using ERA5 on aerodynamic resistance should impact the other bi-directional flux models used in a similar fashion. Although the other modeled resistances may be impacted differently. We previously considered modeling results from the other two bidirectional $NH_3$ exchange models, however we deemed the inter-model comparison worthy of its own publication. A future paper will consider the differences between these models, compare with $NH_3$ fluxes derived using concentration gradient measurements, and improve the parameterization of each model above a forest ecosystem.

Although using a bidirectional air-surface exchange scheme is more theoretically correct than using a traditional big-leaf dry deposition scheme, the former does not necessarily perform better than the latter in the simulated ammonia fluxes on seasonal to annual basis and in regional-scale air-quality modeling, as reported by several existing studies. This is because modelling the bi-directional flux requires additional model parameters such as the soil and canopy NH3 emission potentials, which may not be available at high spatial resolution on the reginal scale. Besides, more model parameters can introduce additional uncertainties. Can you provide any insights on this point with your data and some additional analysis?

We probed the sensitivity of our results to changes in several input parameters, including TAN, LAI and $NH_3$ concentrations. We found that our modelled $NH_3$ fluxes were very sensitive to TAN. The TAN sensitivity analysis is now included in the supplement. For this study, we had soil measurements to pull from. However, those values are not typically available and may be highly spatially variable, as they were in RMNP. Changing the TAN value by one standard deviation, as determined by Stratton et al. (2018), changed the mean $NH_3$ flux by $\pm 0.9$ ng N m$^{-2}$ s$^{-1}$, a large deviation, given the size of typical $NH_3$ fluxes in RMNP. We also found that $NH_3$ fluxes are highly sensitive to $NH_3$ concentration value. In the sensitivity analysis now included in the supplement, increasing the $NH_3$ concentration by 9% increased the annual deposition by 47%. We chose a 9% increase because it is the RMSE determined between passive $NH_3$ measurements and other measurement types by Puchalski et al. (2011). From these results, we demonstrate that bidirectional fluxes are extremely sensitive to chosen parameters, and $NH_3$ concentrations, in addition to time resolution and meteorology datasets.**Citation**: https://doi.org/10.5194/egusphere-2025-1167-EC1

---

## Editor Decision (ED1)

**EGUSPHERE-2025-1167 | Research article**

**Sensitivity of Simulated Ammonia Fluxes in Rocky Mountain National Park to Measurement Time Resolution and Meteorological Inputs**

Naimie et al. (2025)

**Anonymous Referee #3**

**[General comment]**

The authors measured NH3 concentrations with a high time resolution of 30-sec and estimated NH3 flux using a bidirectional exchange model above a subalpine forest, covering a one-year period from September 2021 to August 2022. They proposed that the underestimation of NH3 dry deposition inferred from model using biweekly basis concentrations can be improved by (i) applying a correction factor derived from the relationship with fluxes inferred using 30-min concentration, or (2) estimating the diurnal variation of the biweekly basis concentration.

Observational data on NH3 in forest ecosystems are scarce, and long-term measurements of NH3 concentrations with high temporal resolution are of great value. In addition, given the limited application of bidirectional exchange models in forests compared to grasslands and croplands, this study not only provides valuable information for forests but also proposes a potentially practical approach for improving dry deposition estimation with higher accuracy by utilizing the data of monitoring network such as AMoN.

The concept and overall approach in this study are excellent, and I commend the authors for their efforts for treating a large amount of observation dataset and handling long-term modeling. However, it seems that there is still room for further discussion regarding the specific aspects of the methodology and analysis of both observations and models that support the results and conclusions of this study.

**[Major concerns]**

**1. QA & QC for measured NH3 concentration**

- Upon reviewing the dataset referenced by the authors ("Rocky Mountain National Park Ammonia Data"), I have several concerns regarding how the observational data, which is the foundation of this study, were processed and quality controlled.
- For the biweekly NH3 measurements obtained using passive samplers, the authors mention there were 27 sampling periods, with samples collected in duplicate. From 23 August 2021 to 9 November 2021, four samples were collected per period. However, the number was reduced to two after this period. I think the authors used mean values for each period for analysis, but it is

- unclear whether the authors accounted for the differing number of samples when averaging. Were the sample sizes standardized or weighted in any way?
- Additionally, there is only one valid passive sample for the period from 30 March 2022 to 14 April 2022. Moreover, the sampling period from 26 May 2022 to 21 June 2022 is approximately one month. There are also cases of overlapping sampling periods (e.g., 23 August to 6 September, and 29 August to 13 September 2021). However, the manuscript does not mention how these irregularities were addressed.
- Regarding the high temporal resolution NH3 measurements of AirSentry, the detection limit is 0.070 ppbv. According to the raw data (RMNP\_AirSentry\_NH3\_2021\_2022.csv) from 9 July 2021 to 13 July 2022, approximately 18.6% of the total values fall below the detection limit. Surprisingly, there is even data with a concentration of "0" (approximately 10.8%). It is highly likely that a substantial portion of the data are also below the quantification limit. I am seriously wondering how the authors handled such data, as the authors state that "Only NH3 data missing due to power outages have been removed from the AirSentry dataset".
- While missing data are understandable in long-term field observations, it is essential for the authors to clearly describe their QA/QC procedures: how sub-detection-limit values were treated, how much of the dataset exceeded the detection and quantification limits, and what criteria were used to ensure the reliability of the measurements. Without such information, the credibility of the dataset and the study's conclusions may be undermined.

**2. Uncertainty in the measured NH3 concentration**

- In my opinion, the overall uncertainty of this study depends heavily on the accuracy of the measured NH3 concentrations. The authors also state that "increasing the NH3 concentration by 9% increased the annual deposition by 47%", highlighting the strong sensitivity of model outputs to measured concentrations. While other referees have suggested sensitivity analyses for various parameters of the bidirectional exchange model, the most critical first step is to clarify the uncertainties in the observational data. Addressing these uncertainties would not only enhance the credibility of the measurements but also improve the reliability and interpretability of the model-based analyses. Assuming proper QA/QC procedures were conducted, I offer the following specific suggestions:
- It is generally known that NH3 exhibits strong seasonal and diurnal variability in both flux and concentration, as is evident in Figure 2. Before scaling high-resolution AirSentry data to match passive sampler, a direct comparison of the two measurement techniques should be quantitatively presented. How well do the two measurements agree? If there are systematic differences, is there any evidence of seasonal bias? In my experience, passive samplers requiring multiple lab processing can be prone to contamination, particularly during warmer seasons.

• Although the authors state that "Passive NH3 sampling methods have been shown to have a low bias", the raw data reveal cases where concentrations differ by more than a factor of 2 between passive samplers during same sampling period (e.g., from 10 May to 26 May 2022; 5 July to 18 July 2022; and 18 July to 2 August 2022). Given such large discrepancies in measured concentration using passive samplers, is it truly necessary to scale high resolution data from AirSentry, which is capable of detecting NH3 at levels as low as 0.070 ppbv, by passive sampler? It also seems that Referee 2's concerns regarding this issue have not yet been fully addressed.

**3. Seasonal variation in NH3 concentration and inferred flux**

- Assuming the QA/QC procedures and the uncertainties in the observational data discussed above are adequately addressed, I offer the following suggestion concerning the seasonal variation of NH3 concentration and fluxes. As the author described, "In practice, fluxes can change quickly and even reverse direction with changing environmental conditions." However, the analysis and discussion about the seasonal variation in NH3 concentration and inferred flux appear to be insufficient.
- Although direct measurements data were not presented at the forest of this study, recent longterm observations have reported that NH3 flux in forest forests generally show emission during warm seasons and deposition during other seasons (Melman et al., https://doi.org/10.1016/j.atmosenv.2024.120976). However, the estimated fluxes are quite small level than previous studies ranging from -5 to 5 ng N m-2 s-1, and no large seasonal variation can be seen (as shown in Figure 3). Furthermore, emission and deposition do not alternate in shortterm, but in some cases show consistent deposition, particularly from late July through September in 2022. I can understand the state of "we focused on the impacts of measurement resolution to probe the impacts of time resolution and reanalysis meteorology," however, the lack of detailed discussion on the seasonal variation leaves a gap in the interpretation. As a fundamental step to enhance the methodological reliability of this study, the authors should analyze the seasonal patterns in NH3 flux and concentration more thoroughly, and explore their potential causes, before addressing the annual deposition estimates and discussing the differences arising from the use of various concentration datasets and meteorological inputs.
- The compensation point, which determines both the direction and magnitude of the estimated NH3 flux strongly depends on meteorological factors such as temperature as well as ambient NH3 concentration. Despite a previous suggestion by Referee #1, the meteorological conditions at the study site remain largely undocumented. The revised manuscript includes only mean values for temperature and relative humidity, but it is not even clear over what period these values were calculated. In addition, only qualitative descriptions of rainfall and snowfall are provided. With such limited information, it is difficult to interpret the temporal dynamics of the NH3 flux or to

- assess uncertainties in the annual deposition estimates. At a minimum, the authors should clearly present seasonal variations in temperature, relative humidity, and rainfall, as well as the snowfall periods during the observation period to support their analysis.
- While the authors have conducted sensitivity analyses on parameters such as LAI and TAN in response to referee comments, I think these are relatively minor issues at this stage. The primary parameters that determine the compensation point of  $(\chi_{z0})$  in the model used at this study) are temperature and emission potential of stomata and soil. In addition to temperature, the emission potential, especially for stomata, also varies seasonally (Flechard et al., https://doi.org/10.5194/bg-10-5183-2013). It is reasonable to assume a constant emission potential within a given season; however, for a one-year study such as this, it would be more appropriate to account for seasonal variation. For example, assigning higher values in summer and lower values in winter would better reflect the expected physiological and environmental dynamics. Moreover, as Referee #1 pointed out, the stomatal emission potential used in this study appear significantly lower than those reported in Zhang al. (2010 https://doi.org/10.1029/2009JD013589) or Massad et al. (2010). Sensitivity analysis of the emission potential is a critical step when applying the bidirectional exchange model and should be prioritized. Even if the influence of emission potential on the  $\chi_{z0}$  is limited under low temperature due to the high-altitude site, this feature itself could be an important characteristic worth highlighting. Accordingly, seasonal variation in temperature should be explicitly shown and discussed in this context.
- While there are still many uncertainties regarding soil processes, and I understand the difficulty in handling soil emissions in bidirectional exchange models, existing model such as Zhang et al. (2010) provide useful guidance. For example, the soil emission potential is set to zero during periods with snow cover in this model. I recommend referring to such practices in your analysis and considering whether a similar treatment may be appropriate for your study.

**[Minor and technical comments]**

- Next time, please clearly indicate which parts of the manuscript have been revised in response to
  the referee's comments. For example, you could write: "We revised the sentence (Line: xx-yy in
  the Author's tracked changes version)."Additionally, there are many typographical errors
  throughout the manuscript. Please perform a thorough proofreading before submitting
  your revised version.
- 2. Figure 1: The phrase "at 7.5-arc-second spatial resolution" is difficult to immediately understand. Pleased convert it to "225 m" for clarity.
- 3. 2.1 Site location: According to the aerial photo at the provided coordinates (40.275903, 105.54596), there appear to be several buildings within approximately 100 meters of the NEON

- tower. This suggests that the condition differs from that of a typical forest observation site. Furthermore, the spatial extent shown in Figure 1 is too large to understand the site-specific conditions. Please provide additional information, such as a photograph of the observation tower, a site map illustrating the surrounding environment, or a schematic diagram of the observation.
- 4. Line115 in Author's tracked changes version (same as below): Please correct the unit from "C" to "°C" and insert spaces before and after the "=" sign. Also, clarify the period over which the means of air temperature and relative humidity were calculated. At a minimum, the total precipitation and snowfall events during the observation period should be explicitly stated.
- 5. Line 117: Please specify the height at which these meteorological parameters (e.g., temperature, humidity) were measured. Also, indicate the temporal resolution of the measurements (e.g., 10-minute averages, hourly, etc.).
- 6. Line 119: Please clarify which specific product was used to determine the LAI value. A value of LAI = 0.8 is quite low for a forest site, which may be due to the coarse spatial resolution of 1km grid that includes bare land, and bulling in addition to vegetation. As the study develops, the authors should consider measuring LAI directly around the tower using a canopy analyzer or similar instrument. LAI is a critical parameter for modeling NH3 emissions and deposition. For example, in the model by Zhang et al. (2010), the stomatal emission potential is set to zero when LAI < 0.5. Additionally, LAI is used to parameterize both in-canopy aerodynamic resistance and cuticular resistance. This is also a key issue for the authors' planned model inter-comparison paper and should be addressed carefully.
- 7. Line 121: Please indicate specifically which section or figure of the supplementary information readers should refer to. This comment applies to other instances of vague referencing throughout the manuscript as well.
- 8. Line 136: If I understand correctly, the Monin-Obukhov length (L) also must be calculate in the NEON site, and the methods for calculating L differ between Sections 2.2.1 and 2.2.2. At the NEON site, it seems that L was estimated using the sonic virtual temperature, the covariance of vertical wind and sonic virtual temperature, and the friction velocity derived from the 3D wind components measured by a sonic anemometer (Please specify the manufacturer name and model number of the instrument. Young ? or Gill?).
  - In contrast, the method for calculating L from ERA5 data is unclear in the manuscript. How did you derive the surface buoyancy flux  $(\overline{w'\theta_{v'}})$ 's from ERA5? This typically requires both sensible and latent heat fluxes, along with air density. Note that ERA5 provides both "surface sensible heat flux" and "instantaneous surface sensible heat flux"; please specify which was used. To enhance transparency and allow others to reproduce your method, I strongly recommend including the equations used (e.g.,  $L_{\text{sonic}} = ...$ , and  $L_{\text{ERA5}} = ...$ ), and a table listing the specific variables used in each case. This would also help readers understand the discrepancies between

 $L_{sonic}$  and  $L_{ERA5}$  that are discussed in the supplemental information. Furthermore, since ERA5 has a 1-h temporal resolution, how was this data used with 30-min concentration data for flux calculation?

The phrase "Eq.(5.7c) from Stull (1988)" is difficult to follow. I suggest rephrasing it as: "... was calculated using Eq. (1) following Stull (1988)".

- Line 146: Please provide the manufacturer name and model number instead of listing only a URL.
   The same request applies to other instruments mentioned in the manuscript, including the 3D sonic anemometer, ion chromatography system, and AirSentry.
- 10. Line 151-152: There are several types of denuders (e.g., annular, multi-channel, honeycomb), but I am not familiar with the term "University Research Glassware Denuders". Is this a proper noun or brand name? Please clarify or revise how this is described in the manuscript.
  - And I do not agree with that the passive sampler have a low bias based on your low data.
- 11. Line 157: "Boulder, CO" may refer only to a location and does not specify the manufacturer or model. The correct company name is "Particle Measuring Systems." Please revise accordingly.
- 12. What does "1/4" means? And please correct the unit from "C" to "°C".
- 13. Line159: Did you perform any tests to assess potential NH3 losses within the sampling tube? Given that NH3 is highly soluble in water, it may be adsorbed onto the tube walls under high humidity conditions.
- 14. Line: 168: Please specify "the effect of NH3 ..." on what? The sentence is currently ambiguous.
- 15. Line: 169: The expression "30-min frequency" is potentially misleading. Since AirSentry's time resolution is already expressed as measured by "30-sec frequency", the 30-minute value generated here should be rephrase.
- 16. Figure 2: The labels "2021" and "2022" on the x-axis only need to appear once at the beginning of each year. Instead, it would be more helpful for readers to better illustrate seasonal trends if the figure included month labels across the full period.
- 17. Line 180: If my understanding is correct, the data count would exceed 3,000 if values below the detection limit or those reported as zero are also included.
- 18. Line 190: Please correct the formatting of "Fig (S1)" to "Fig. S1".
- 19. Line 202: Please specify what type of wet deposition data were used.
- 20. Line 203: Please use the same format for latitude and longitude notation as used for other sites to ensure consistency throughout the manuscript.
- 21. Line 210: The phrase "dry deposition is generated" sounds unnatural. Dry deposition is typically inferred or estimated, not "generated." Please revise this expression accordingly. The same applies to the use of "generation" in reference to  $V_d$ .
- 22. Line 214: The abbreviation "(Vd)" should be introduced at Line 211 when "deposition velocity" is first mentioned.

- 23. Line 215: Why "dry deposition velocity" is being used at here instead of Vd.
- 24. Line 217: Although I understand this is based on previously studies, please note that the Vd of NH3 is not so easily defined. This may partially explain the discrepancy in annual deposition amount discussed at Line 365-368.
- 25. Line 219: The bidirectional flux model by Massad et al. (2010) is more complex than other models used in previous studies, and the calculation of theχz0 is difficult. This is why the authors are required to conduct various sensitivity analyses. It would be better to describe the advantages behind selecting this model.
- 26. Line 222: If I understand correctly, the authors appear to have misunderstood the framework of the model of Massad et al. (2010). In this model, the direction of total flux is determined by the difference between atmospheric concentration ( $\chi_a$ ) and  $\chi_{z0}$ . The canopy compensation point ( $\chi_c$ ) is an intermediate parameter for determining  $\chi_{z0}$ , not the determinant of direction of total flux. This distinction is important because the model differs conceptually from simpler models. Please revise the explanation to ensure accuracy.
- 27. Figure 3: The term "surface compensation point" is incorrect; as far as I know, this terminology is not used in previous literature.  $\chi_{z0}$  is the compensation point at height of  $(d + z_0)$ . Please correct "Stomata" to "Stomatal", and "cuticle" to "cuticular". Also, "laminar" is no need for  $R_{bg}$ .
- 28. Line 235: Is Figure 3 showing "relationship" rather than a conceptual diagram? Please clarify and revise the description accordingly.
- 29. Line 239: Obukhov length, displacement and roughness length have already been defined as L, d, and z0.
- 30. Line 245: Please delete "captures the aerodynamic resistance from within the canopy layer and". While this may apply to Rac, it is incorrect in the context of Rg.
- 31. Line 247: The phrase "using Eq. 16 and Eq. 17 from equations of Massad et al. (2010)." may confuse readers, especially since the present manuscript also contains Eq.(16) and Eq.(17).
- 32. Line 252: Could you elaborate on how the parameter Rbg was determined? This parameter are not widely compiled, and your approach would provide valuable information for future studies.
- 33. Line 257: According to Table 1 in Zhang et al. (2003 https://doi.org/10.5194/acp-3-2067-2003), I could not find a rstmin value of "225" for any land use category. Is the mistake of 250? Also, did you assign Rst at nighttime to be infinite value considering the stomatal closure following Zhang et al. (2003)? This is a critical assumption in modeling bidirectional exchange of NH3.
- 34. Line258: Based on your response to Referee #1, it seems there may be a misunderstanding regarding Rw. As I understand it, this is an empirical formulation that accounts for four different vegetation types, not only for "Douglas Fir." Moreover, the effect of LAI and Temperature is

already considered at this empirical formula. It is important to avoid applying such models blindly without a full understanding of their basis.

As an additional point, changing the cuticular resistance formulation can also significantly affect NH3 fluxes in addition to the emission potential. I recommend considering the case study of Xu et al. (2023 https://doi.org/10.1016/j.atmosenv.2023.120144) using a bidirectional exchange model in your uncertainty discussion or future sensitivity analysis.

- 35. Line 262: This is self-evident from Eq.(8), and Eq.(9) is unnecessary. Is there any case in which relative humidity exceeded 100% at the NEON site?
- 36. Line 263: Although the exclusion of HCl likely has minimal influence on the results, was there a specific reason it was not considered in the calculation? I suspect that the higher acidity ratio (AR) observed in winter may be due to extremely low NH3 concentrations. Is it possible that a higher AR facilitated NH3 deposition in the site? According to the study by Xu et al. (2023), Rw can be highly sensitive to this factor.
- 37. Line 269: The stomatal and ground compensation points have already been defined as  $\chi_s$  and  $\chi_g$ .
- 38. Line 270: Why did you not use the formula of Massad et al. (2010), which calculates  $\Gamma_s$  for Unmanaged site based on nitrogen input? The  $\Gamma_s$  value calculated by this formula is about 10 times larger than the value used in this study. Do you expect this have any impact on the flux calculations?
- 39. Line 272: What exactly do you mean by "ratios" in this context?
- 40. Line 275: Same comment with comment 39 for  $\chi_s$  and  $\chi_g$ . Also, "Eq. (3) and Eq. (5) of Stratton et al. (2018)" could confuse readers, as the equation numbers overlap with those in your manuscript.
- 41. Line 281: The authors described "ground compensation points were calculated according to Massad et al. (2010)" in Line 269. However, they also described soil compensation point was calculated according to Stratton et al. (2018) in Line 275, and  $\chi_g$  is not calculated follow the form of Eq. (11). If this is the mistake of  $\Gamma_g$ , I can understand. Which is correct?
- 42. Line 286: Same comment with 40 for  $\chi_c$ .
- 43. Line 287:χa was already defined and explained earlier in Line 222.
- 44. Line 291: Same comment with 43 for d and z0. Also, note that it is conventional to use lowercase z, not uppercase Z. Please revise accordingly.
- 45. Line 296: The explanation of inferred flux here is scientifically incorrect. It raises concerns about whether the authors even fully understand the resistance model framework. Also, "roughness height" is incorrect as already been pointed out by Referee #1.
- 46. Line 304: Please correct the section title to "Simulated bidirectional exchange flux of NH3".

- 47. Line 307-309: This explanation has already been presented in the Methods section. And I cannot agree using the word of "relative magnitudes". Again, "Surface compensation point" is not an appropriate term; please revise all instances in the manuscript.
  I suggest adding a plot of the "difference" between χa and χz0 in Figure 4. This "difference" directly determines the direction and magnitude of the flux and would provide valuable insight into seasonal variation.
- 48. Line 313: For clarity, please define the seasons as used in your study (e.g., spring = March to May). From Figure 4, it is not clear how "The largest periods of net emission occur in the spring" was concluded. It appears that the largest emissions occur from late June to July. Why are large depositions observed before and after this period (even in same summer)? What factors do you think are influencing these patterns?
- 49. Line 321, 323: Same comment with 28.
- 50. Line 324: Based on Figure 5, it is difficult to support the claim that "Winter periods of net emission (see Fig. 4b) are driven by the ground flux." If these fluxes exhibit seasonal variation, boxplots may obscure such information, a time series plot would be more appropriate to reveal these dynamics because total flux is sum of these fluxes. Furthermore, as previously mentioned, some models assume no soil emission under snow-covered conditions. Then, why is soil emission estimated to be larger in winter when temperatures are low, and snow is present?
- 51. Line 325: Do you have any hypotheses or supporting information on how snow cover affects the flux?
- 52. Line 338: The term "surface exchange" may be more appropriate than "dry deposition" here.
- 53. Line 346: The phrasing should be revised to "we also observe peak deposition fluxes" for clarity.
- 54. Figure 6: While the discussion of diurnal variation is valuable, analyzing the full annual dataset may obscure seasonal characteristics. Do the observed diurnal patterns hold true across all seasons, for example, during winter when concentrations and air temperature are lower? Are stronger daytime emissions observed during summer?
- 55. Line 365-368: Is it also possible that the applied bidirectional flux model underestimates the actual amount of dry deposition? This relates to the previous question: why is dry deposition substantially higher in May and August compared to other months? Could the low dry deposition in June and July be due to large daytime emissions? Since wet deposition also become lower during these months, did less rainfall favored NH3 emission?
- 56. Line 379: According to Figure 7(b), the largest NH3 dry deposition appear in "May" and August. Please revise the text accordingly.
- 57. Line 380-382: Can you explain why the proportion of reduced nitrogen is so high in this area, in relation to the transport and sources mentioned above?

- 58. Line 389: Please be consistent in terminology. Use either "time resolution" or "time-resolution" throughout the manuscript, but do not mix both styles.
- 59. Line 406: Please consider expanding your discussion on the reasons for the differences in modeled flux. If I understand correctly, one key factor may be the diurnal variation in the flux: At high temporal resolution, models usually reproduce large daytime NH3 emissions driven by increasing temperature. At night, stomatal closure and reduced turbulence lead to larger stomatal and in-canopy aerodynamic resistances, suppressing emissions from both stomata and ground. Simultaneously, elevated relative humidity and acidity ratio (AR) enhance deposition. Therefore, deposition at nighttime may largely contribute to the annual dry deposition in 30-min resolution. However, at lower temporal resolution, these diurnal dynamics are averaged out, potentially leading to overestimation of emission and underestimation of deposition.
- 60. Figure 10: While the overall trends of deposition and emission appear roughly consistent, there are notable differences in the magnitude of deposition fluxes in some cases. Both fluxes seem to exhibit systematic bias. What could be causing this? It is also puzzling that the annual dry deposition totals are same despite these differences.
- 61. Line 438: Does this mean that you applied a monthly diel pattern of each month? If so, this sentence is an insufficient explanation.
- 62. Line 443: Please revise "Dry deposition inferential" to "Bidirectional exchange".
- 63. Line 446: Again, how did you simulate 30-min fluxes using ERA5 reanalysis data with 1-hour temporal resolution?
- 64. Line 451-452: This sentence is unclear and please rephrase. Figure 11 suggests that the ERA5-based flux show smaller emission compared to the NEON simulation, which could result in the higher dry deposition amount.
- 65. Figure 11: The title appears redundant since it merely repeats the caption.

  It may be more informative to plot by specifying season or by day/night. This could help identify reasons for the discrepancies more clearly.
- 66. Line 460: Same comment with 44 for  $R_a$  and  $\chi_{z0}$ .
- 67. Line 472: Same comment with 66 for u\* and L.
- 68. Line 480: Same comment with 68 for L.

  If the u\* from ERA5 were corrected using NEON, would the simulation results become more consistent between the two datasets?
- 69. Line 481-482: Do the observed differences in Ra directly cause the discrepancies in dry deposition amount? If so, please describe. Also, it is unclear why χz0 values are consistent despite significant differences in Ra (in Line 461-462).

- Moreover, Figure S2(h.) shows a large discrepancy in RH ( $R^2 = 0.34$ ), and ERA5 values being substantially higher than NEON. As RH is a critical input for  $R_w$ , how does this discrepancy influence fluxes? Why was this not discussed in the main manuscript?
- 70. Line 485: Considering the editor's comment, the phrase "best simulated" may be inappropriate at this stage.
- 71. Line 489: As noted previously, please use consistent terminology throughout the manuscript: either "bidirectional" or "bi-directional," but not both.
- 72. Line 510-514: This methodology is quite promising. If you could briefly describe its potential applicability to other sites or its utility for future research, this would enhance the academic contribution of this study.
- 73. Figure S7: The meaning of "calculated fractional differences" is unclear. What can be seen from this figure is that the magnitude of flux in response to the  $\chi_a$  scaling factor differ considerably by season, regardless of the two time periods.

---

## Author Response (AR2)

We would like to thank Referee #3 for their thorough review of all materials and the data repository. We appreciate their insights and have responded to each concern. Our response to the Referee is included in RED.

**We have also added the following section to the results to better address the editor's comments.**

"In this analysis, we simulated the bidirectional exchange of NH3 above a forest ecosystem using the model proposed in Massad et al. (2010). However, there are other bidirectional exchange models (e.g., Zhang et al., 2010; Pleim et al., 2013) and their simulated fluxes may differ significantly from the model used here (Jongenelen et al., 2025). In the bidirectional exchange model used here, we observe that the selected inputs for NH3 concentration and meteorological data may introduce biases into the simulated NH3 fluxes. This may also be true for the other models when simulating NH3 bidirectional exchange, a good topic for future research."

Jongenelen, T., M. van Zanten, E. Dammers, R. Wichink Kruit, A. Hensen, L. Geers, and J. W. Erisman (2025), Validation and uncertainty quantification of three state-of-the-art ammonia surface exchange schemes using NH3 flux measurements in a dune ecosystem, *Atmos. Chem. Phys.*, *25*(9), 4943-4963, doi:10.5194/acp-25-4943-2025.

**EGUSPHERE-2025-1167 | Research article**

Sensitivity of Simulated Ammonia Fluxes in Rocky Mountain National Park to Measurement Time Resolution and Meteorological Inputs

**Naimie et al. (2025)**

**Anonymous Referee #3**

**[General comment]**

The authors measured NH3 concentrations with a high time resolution of 30-sec and estimated NH3 flux using a bidirectional exchange model above a subalpine forest, covering a one-year period from September 2021 to August 2022. They proposed that the underestimation of NH3 dry deposition inferred from model using biweekly basis concentrations can be improved by (i) applying a correction factor derived from the relationship with fluxes inferred using 30-min concentration, or (2) estimating the diurnal variation of the biweekly basis concentration.

Observational data on NH3 in forest ecosystems are scarce, and long-term measurements of NH3 concentrations with high temporal resolution are of great value. In addition, given the limited application of bidirectional exchange models in forests compared to grasslands and croplands, this study not only provides valuable information for forests but also proposes a potentially practical approach for improving dry deposition estimation with higher accuracy by utilizing the data of monitoring network such as AMoN.

The concept and overall approach in this study are excellent, and I commend the authors for their efforts for treating a large amount of observation dataset and handling long-term modeling. However, it seems that there is still room for further discussion regarding the specific aspects of the methodology and analysis of both observations and models that support the results and conclusions of this study.

**[Major concerns]**

**QA & QC for measured NH3 concentration**

Upon reviewing the dataset referenced by the authors ("Rocky Mountain National Park Ammonia Data"), I have several concerns regarding how the observational data, which is the foundation of this study, were processed and quality controlled.

For the biweekly NH3 measurements obtained using passive samplers, the authors mention there were 27 sampling periods, with samples collected in duplicate. From 23 August 2021 to 9 November 2021, four samples were collected per period. However, the number was reduced to two after this period. I think the authors used mean values for each period for analysis, but it is samples when averaging. Were the sample sizes standardized or weighted in any way?

We apologize for the confusion in regard to this sampling. From Aug. 23 to Nov. 9, two samples were deployed at the tower top, and two were deployed lower on the tower. After this period, passive samples were only deployed at the tower top. Only the tower top data was used for this analysis. You are correct that we used a mean value for each period. This sentence has been added to the passive measurement methods section: "Due to site access issues, some samples had durations longer than 2 weeks. To create a consistent dataset, all data were aggregated to a 2-week average. In the case where two samples overlapped a 2-week period, they were combined using a weighted average. One sample was below the detection limit and was removed from this analysis."

Additionally, there is only one valid passive sample for the period from 30 March 2022 to 14 April 2022. Moreover, the sampling period from 26 May 2022 to 21 June 2022 is approximately one month. There are also cases of overlapping sampling periods (e.g., 23 August to 6 September, and 29 August to 13 September 2021). However, the manuscript does not mention how these irregularities were addressed.

Thank you for taking such a detailed look at the raw data used. We have added text to explain how the overlapping periods were handled (see above). We have also added the following text to make other sampling discrepancies clearer to the reader: "During the sampling period, some sampling periods were longer than others due to issues accessing the site. To compare periods of a consistent length, all data was taken to a 2-week average using a weighted mean of the data available during that period. One sample was below the detection limit and removed from this analysis." This text was added to lines 154-156.

Regarding the high temporal resolution NH3 measurements of AirSentry, the detection limit is 0.070 ppbv. According to the raw data (RMNP\_AirSentry\_NH3\_2021\_2022.csv) from 9 July 2021 to 13 July 2022, approximately 18.6% of the total values fall below the detection limit. Surprisingly, there is even data with a concentration of "0" (approximately 10.8%). It is highly likely that a substantial portion of the data are also below the quantification limit. I am seriously wondering how the authors handled such data, as the authors state that "Only NH3 data missing due to power outages have been removed from the AirSentry dataset".

Thank you for the detailed look at the data repository. There are a few updates we can make to help the reader understand the data processing. First, the data with a concentration of "0" were removed in our analysis, but were not replaced for the data upload. This error has been fixed, and the data repository does not contain any values of zero. Second, what is currently

included in the manuscript for a detection limit (70 ppt) is for the 30-second measurements. The data used is a 30-minute average of these measurements. Averaging 60 data points increases the signal-to-noise ratio and lowers our detection limit. The following text has been added to the methods section in regards to the AirSentry data (lines 168-172): "The limit of detection for 30-second measurements 70 pptv. For this data analysis, NH3 concentration data was averaged to 30-minute mean values. Averaging data points increases the signal-to-noise ratio. We approximate that the signal-to-noise ratio increases proportionally to the square root of the number of samples (n = 60) (Dempster, 2001). In this case, the signal-to-noise ratio increases by a factor of 7.7, reducing the limit of detection to 9 pptv for 30-minute mean NH3 concentrations. Across the year of data collection, 101 points fell below the detection limit." A description of this detection limit should be helpful to the reader. Later in the text (lines 190-192), we have added the following line to explain why those 101 points were allowed to remain in the dataset: "The 101 30-minute average NH3 concentration values below the AirSentry detection limit, representing 0.5% of the total measurement period, were assumed to represent a random distribution below the detection limit and retained for post-process scaling from the passive observations." Additionally, the mean of values below the detection limit is 0.005 ppbv NH3, which is approximately ½ of the MDL.

While missing data are understandable in long-term field observations, it is essential for the authors to clearly describe their QA/QC procedures: how sub-detection-limit values were treated, how much of the dataset exceeded the detection and quantification limits, and what criteria were used to ensure the reliability of the measurements. Without such information, the credibility of the dataset and the study's conclusions may be undermined.

We appreciate the feedback and encouragement to include more description of the QA/QC procedures performed on the dataset. We have updated the methods section, as described above, to include more discussion of QA/QC. We hope this will help the reader see the credibility of the datasets and our conclusions more clearly.

**Uncertainty in the measured NH3 concentration**

In my opinion, the overall uncertainty of this study depends heavily on the accuracy of the measured NH3 concentrations. The authors also state that "increasing the NH3 concentration by 9% increased the annual deposition by 47%", highlighting the strong sensitivity of model outputs to measured concentrations. While other referees have suggested sensitivity analyses for various parameters of the bidirectional exchange model, the most critical first step is to clarify the uncertainties in the observational data. Addressing these uncertainties would not only enhance the credibility of the measurements but also improve the reliability and interpretability of the model-based analyses. Assuming proper QA/QC procedures were conducted, I offer the following specific suggestions:

This is an important point, and part of our hopeful takeaway for readers is that NH3 concentration measurements have an important and highly variable impact on simulated NH3 fluxes. In the sections above, we have addressed the QA/QC concerns discussed. This will hopefully aid the reader in understanding the reliability of our data.

• It is generally known that NH3 exhibits strong seasonal and diurnal variability in both flux and concentration, as is evident in Figure 2. Before scaling high-resolution AirSentry data to match passive sampler, a direct comparison of the two measurement

techniques should be quantitatively presented. How well do the two measurements agree? If there are systematic differences, is there any evidence of seasonal bias? In my experience, passive samplers requiring multiple lab processing can be prone to contamination, particularly during warmer seasons.

A direct comparison of the mean biweekly passive NH3 concentrations and AirSentry data has been added to Fig. S1 in the supplementary information. We have also added the following text: "AirSentry NH3 concentration measurements at the nearby EPA shelter are compared with mean biweekly passive NH3 data. The biweekly passive NH3 values are a weighted average of all tower-top passive measurements made during the 2-week period. Note, the AirSentry NH3 measurements were taken at the nearby EPA shelter, at a height of 2 m above a grassland ecosystem, and the passive NH3 measurements were made at a height of 25 m above a forest ecosystem on the NEON tower." A linear fit to the datasets departs from the 1 to 1 line (slope = 1.07, intercept = 0.1 ppbv,  $R^2 = 0.71$ ), but as noted, the site differences make it challenging to separate systematic biases from measurement techniques and potential concentration differences between the two sites. We do not observe a seasonal variation in the difference between measurement techniques.

• Although the authors state that "Passive NH3 sampling methods have been shown to have a low bias", the raw data reveal cases where concentrations differ by more than a factor of 2 between passive samplers during same sampling period (e.g., from 10 May to 26 May 2022; 5 July to 18 July 2022; and 18 July to 2 August 2022). Given such large discrepancies in measured concentration using passive samplers, is it truly necessary to scale high resolution data from AirSentry, which is capable of detecting NH3 at levels as low as 0.070 ppby, by passive sampler? It also seems that Referee 2's concerns regarding this issue have not yet been fully addressed.

Our reason for scaling the AirSentry data is to avoid differences between atmospheric NH3 concentration above a grassland, where the AirSentry was placed, and NH3 concentration above a forest ecosystem, where the other measurements were made. We also scale the measurements to ensure that the 2-week mean value is consistent for NH3 flux simulations with high time resolution NH3 concentrations and the biweekly NH3 concentration data. Due to the different locations of these measurements, we did not directly assess any biases in the passive NH3 data. We use previous works to estimate the potential low bias, and use it as a bound for how NH3 fluxes could change if such a bias was observed. Given our data availability, we had no way to assess how the passive measurements would have compared with AirSentry NH3 measurements made at the same location.

**Seasonal variation in NH3 concentration and inferred flux**

• Assuming the QA/QC procedures and the uncertainties in the observational data discussed above are adequately addressed, I offer the following suggestion concerning the seasonal variation of NH3 concentration and fluxes. As the author described, "In practice, fluxes can change quickly and even reverse direction with changing environmental conditions." However, the analysis and discussion about the seasonal variation in NH3 concentration and inferred flux appear to be insufficient.

We appreciate the discussion provided here by Referee #3. We have addressed the QA/QC concerns above, and here will address the direct comments on the seasonality of NH3 and the simulated fluxes.

 Although direct measurements data were not presented at the forest of this study, recent long-term observations have reported that NH3 flux in forest forests generally show emission during warm seasons and deposition during other seasons (Melman et al., 2025 https://doi.org/10.1016/j.atmosenv.2024.120976). However, the estimated fluxes are quite small level than previous studies ranging from -5 to 5 ng N m-2 s-1, and no large seasonal variation can be seen (as shown in Figure 3). Furthermore, emission and deposition do not alternate in short-term, but in some cases show consistent deposition, particularly from late July through September in 2022. I can understand the state of "we focused on the impacts of measurement resolution to probe the impacts of time resolution and reanalysis meteorology," however, the lack of detailed discussion on the seasonal variation leaves a gap in the interpretation. As a fundamental step to enhance the methodological reliability of this study, the authors should analyze the seasonal patterns in NH3 flux and concentration more thoroughly, and explore their potential causes, before addressing the annual deposition estimates and discussing the differences arising from the use of various concentration datasets and meteorological inputs.

Yes, we were also surprised to not have a more clear large seasonal variation in NH3 fluxes. Previous work in RMNP (Pan et al., 2021) also observed deposition fluxes during the summer This conceptually makes sense, because NH3 concentrations are typically higher in the warm seasons. Some previous works may have found deposition during cool seasons in part due to their handling of snow cover. Later in this document and in the general text, we have added a small analysis to understand how shutting off soil emissions in the winter time would have affected our simulated fluxes. We have added some additional discussion of the seasonality of meteorology, to help the reader understand the seasonal cycle in RMNP. We appreciate the referee's feedback that the text is lacking a seasonal interpretation. For an updated annual Nr deposition budget in RMNP and to understand bidirectional flux simulations of NH3 this would be a great focus for future research.

• The compensation point, which determines both the direction and magnitude of the estimated NH3 flux strongly depends on meteorological factors such as temperature as well as ambient NH3 concentration. Despite a previous suggestion by Referee #1, the meteorological conditions at the study site remain largely undocumented. The revised manuscript includes only mean values for temperature and relative humidity, but it is not even clear over what period these values were calculated. In addition, only qualitative descriptions of rainfall and snowfall are provided. With such limited information, it is difficult to interpret the temporal dynamics of the NH3 flux or to assess uncertainties in the annual deposition estimates. At a minimum, the authors should clearly present seasonal variations in temperature, relative humidity, and rainfall, as well as the snowfall periods during the observation period to support their analysis.

We appreciate the Reviewer's feedback on increasing the meteorological information included in the manuscript. We have added seasonal mean values for both RH and temperature to help the reader understand the meteorological conditions at the NEON tower in RMNP. The following text has been added: "Mean values were calculated from September

2021 to September 2022. Snowfall typically occurs between October and May. The seasonal mean temperatures (relative humidities) are as follows: winter (December, January, and February) mean is -3 °C (30%), spring (March, April, and May) mean is 2 °C (44%), the summer (June, July, and August) mean is 15 °C (49%), and the fall (September, October, and November) mean is 8 °C (37%). Precipitation is measured at 1-minute resolution by a Belfort AEPG II 600M weighing gauge. Precipitation events were defined as periods of rainfall separated by at least one hour without precipitation. During our study period, there were 27 precipitation events in the winter, 62 in the spring, 63 in the summer, and 26 in the fall."

While the authors have conducted sensitivity analyses on parameters such as LAI and TAN in response to referee comments, I think these are relatively minor issues at this stage. The primary parameters that determine the compensation point of  $(\gamma z0)$  in the model used at this study) are temperature and emission potential of stomata and soil. In addition to temperature, the emission potential, especially for stomata, also varies seasonally (Flechard et al., 2013 https://doi.org/10.5194/bg-10-5183-2013). It is reasonable to assume a constant emission potential within a given season; however, for a one-year study such as this, it would be more appropriate to account for seasonal variation. For example, assigning higher values in summer and lower values in winter would better reflect the expected physiological and environmental dynamics. Moreover, as Referee #1 pointed out, the stomatal emission potential used in this study appear significantly lower than those reported in Zhang et al. (2010 https://doi.org/10.1029/2009JD013589) or Massad et al. (2010). Sensitivity analysis of the emission potential is a critical step when applying the bidirectional exchange model and should be prioritized. Even if the influence of emission potential on the  $\chi$ z0 is limited under low temperature due to the high-altitude site, this feature itself could be an important characteristic worth highlighting. Accordingly, seasonal variation in temperature should be explicitly shown and discussed in this context.

We tested the sensitivity to TAN, LAI, and NH3 concentration at the suggestion of other referee comments, as noted here. For TAN and LAI, we had good data to make estimates of the reasonable bounds and found the NH3 fluxes to be sensitive to both parameters. Although fluxes are sensitive to temperature, we have temperature data from the NEON tower with very low uncertainty. The equations used do not include an emission potential of soil. Instead, in Eq. (12), the ground compensation point is simulated using TAN. In effect, conducting a sensitivity analysis on TAN determined the sensitivity of the ground compensation point and the relative role of the ground (including soil) on NH3 fluxes. For stomatal emission potential, we used direct measurements of foliage around the NEON tower. The collection process and calculation are described in section S7 of the supplementary materials. This value is lower than those reported by Zhang et al. (2010) and Massad et al. (2010), however, it is directly reflective of our measurements at the site. The stomatal emission potentials summarized in Zhang et al. (2010), and Massad et al. (2010) are highly variable, and there is very limited direct data for seasonal changes above this land surface type. Foliage samples were collected across seasons; however we did not observe a seasonal pattern due to the high sample variability. We probed this sensitivity by running simulations with different values for stomatal emission potential. Doubling the stomatal emission potential decreased the annual deposition flux by 14%. Halving the stomatal emission potential increased the annual flux by 7%.

• While there are still many uncertainties regarding soil processes, and I understand the difficulty in handling soil emissions in bidirectional exchange models, existing model such as Zhang et al. (2010) provide useful guidance. For example, the soil emission potential is set to zero during periods with snow cover in this model. I recommend referring to such practices in your analysis and considering whether a similar treatment may be appropriate for your study.

For this work, we chose to focus on one bidirectional exchange model (Massad et al. 2010) and deeply investigate the impacts of changing the time resolution of meteorological and NH3 concentrations.

We conducted another sensitivity analysis to probe the impact of shutting off soil emissions. For this test, we set  $\chi_g$  equal to zero during the winter (December, January, and February). This changed the wintertime net fluxes from emission to deposition. This sentence was added to the main text "To probe the impact of snow cover, a sensitivity test was conducted setting  $\chi_g$  equal to zero during the winter (December, January, and February), which increased annual deposition by 0.06 kg N ha-1 yr-1. However, this analysis does not take into account how the surface differences may change NH3 fluxes above snow. "This text was added to the supplementary information "Lastly, we tested the sensitivity of flux simulations to the ground compensation point during the winter (December, January, and February), to probe the potential impact of snow cover. In the winter, we set  $\chi_g$  to zero to stop ground emissions. Setting  $\chi_g$  to zero during winter changed the net wintertime flux from emission to deposition and increased the annual NH3 deposition by 0.06 kg N ha-1."

**[Minor and technical comments]**

1. Next time, please clearly indicate which parts of the manuscript have been revised in response to the referee's comments. For example, you could write: "We revised the sentence (Line: xx-yy in the Author's tracked changes version)."Additionally, there are many typographical errors throughout the manuscript. Please perform a thorough proofreading before submitting your revised version.

In the future, and for responses to this review, we will include line-by-line notation for all changes made. We apologize for missing the noted typographical errors noted here and appreciate the reviewer for catching them.

2. Figure 1: The phrase "at 7.5-arc-second spatial resolution" is difficult to immediately understand. Pleased convert it to "225 m" for clarity.

Although 225 m is approximately correct, the USGS outputs this data in arc-seconds because it is generated using latitude and longitude, which are not a consistent distance across the globe. We have changed the text to "at 7.5-arc-second spatial resolution, or approximately 225 meters".

3. Site location: According to the aerial photo at the provided coordinates (40.275903, - 105.54596), there appear to be several buildings within approximately 100 meters of the NEON tower. This suggests that the condition differs from that of a typical forest observation site. Furthermore, the spatial extent shown in Figure 1 is too large to understand the site-specific conditions. Please provide additional information, such as a photograph of the observation tower, a site map illustrating the surrounding environment, or a schematic diagram of the observation.

There are a few small measurement shelters used by NEON within 100 meters of the site, and one larger building ~150 m away. Supplementary Fig. S5 shows the site configuration in more detail, with the LAI values plotted. In Fig. S5 you can see the larger building mentioned, and its distance from the measurement site. The larger map is shown in Fig. 1 to give the reader a sense of the general surroundings and source regions near the site.

4. Line115 in Author's tracked changes version (same as below): Please correct the unit from "C" to "°C" and insert spaces before and after the "=" sign. Also, clarify the period over which the means of air temperature and relative humidity were calculated. At a minimum, the total precipitation and snowfall events during the observation period should be explicitly stated.

In line 115, "6 C" has been updated to "6 °C" and spaces have been added before and after the equals ("=") sign. The following text was added to line 116: "Mean values were calculated from September 2021 to September 2022." We have added the following text: "Precipitation is measured at 1-minute resolution by a Belfort AEPG II 600M weighing gauge. Precipitation events were defined as periods of rainfall separated by at least one hour without precipitation. During our study period, there were 27 precipitation events in the winter, 62 in the spring, 63 in the summer, and 26 in the fall" to describe the precipitation events observed at the RMNP NEON tower.

5. Line 117: Please specify the height at which these meteorological parameters (e.g., temperature, humidity) were measured. Also, indicate the temporal resolution of the measurements (e.g., 10-minute averages, hourly, etc.).

The following sentences have been added to line 117: "The meteorological observations used from the NEON tower are 30-minute mean values. Direct measurements of wind vectors, air temperature, short wave radiation, relative humidity, air density, and air pressure were used from the tower-top measurements (25 m-agl). 3D wind vectors were measured at 20 Hz using the CSAT-3 sonic anemometer (Campbell Scientific Inc., Logan, Utah, USA)."

6. Line 119: Please clarify which specific product was used to determine the LAI value. A value of LAI = 0.8 is quite low for a forest site, which may be due to the coarse spatial resolution of 1km grid that includes bare land, and bulling in addition to vegetation. As the study develops, the authors should consider measuring LAI directly around the tower using a canopy analyzer or similar instrument. LAI is a critical parameter for modeling NH3 emissions and deposition. For example, in the model by Zhang et al. (2010), the stomatal emission potential is set to zero when LAI < 0.5. Additionally, LAI is used to parameterize both in-canopy aerodynamic resistance and cuticular resistance. This is also a key issue for the authors' planned model intercomparison paper and should be addressed carefully.

Thank you for catching this. The LAI product from NEON, detailed in the supplementary information, is a spectroscopic measurement around the NEON site. The following text: "Leaf area index (LAI) is estimated at the site using remotely sensed data at 1 km resolution" has been updated to "Leaf area index (LAI) is estimated at the site using remotely sensed data". The data is not at 1 km resolution, as you can see in Fig. S5. The tiles are output for square km areas. The spatial area shown to generate 0.8 for an LAI is shown in Fig. S5. The area was specifically selected to not include the nearby building or cleared land area. This data was also used to generate the minimum and maximum LAI values for sensitivity testing.

We appreciate the reviewer's input for upcoming papers and will seek to carefully address the impacts of LAI on intermodal comparison and parameterizations.

7. Line 121: Please indicate specifically which section or figure of the supplementary information readers should refer to. This comment applies to other instances of vague referencing throughout the manuscript as well.

To more clearly reference the supplementary information, the following text: "The square kilometer of leaf area index values surrounding the tower site is shown in the supplementary information" has been updated to "The square kilometer of leaf area index values surrounding the tower site is shown in Fig. S5". The following sentence: "The sensitivity to LAI can also be found in the supplementary information", has also been updated to: "The sensitivity to LAI can also be found in section 5 of the supplementary information".

8. Line 136: If I understand correctly, the Monin-Obukhov length (L) also must be calculated in the NEON site, and the methods for calculating L differ between Sections 2.2.1 and 2.2.2. At the NEON site, it seems that L was estimated using the sonic virtual temperature, the covariance of vertical wind and sonic virtual temperature, and the friction velocity derived from the 3D wind components measured by a sonic anemometer (Please specify the manufacturer name and model number of the instrument. Young? or Gill?). In contrast, the method for calculating L from ERA5 data is unclear in the manuscript. How did you derive the surface buoyancy flux  $(w'\theta v^{\overline{7}})$ s from ERA5? This typically requires both sensible and latent heat fluxes, along with air density. Note that ERA5 provides both "surface sensible heat flux" and "instantaneous surface sensible heat flux"; please specify which was used. To enhance transparency and allow others to reproduce your method, I strongly recommend including the equations used (e.g., Lsonic = ..., and LERA5 = ...), and a table listing the specific variables used in each case. This would also help readers understand the discrepancies between Lsonic and LERA5 that are discussed in the supplemental information. Furthermore, since ERA5 has a 1-h temporal resolution, how was this data used with 30-min concentration data for flux calculation?

Yes, you are correct, NEON uses the Monin-Obukhov similarity theory to calculate L using friction velocity (from 3D sonic anemometer), air temperature, and Eddy covariance of wind and temperature. The 3D wind component instrument is now included in the NEON description with this text: "3D wind vectors were measured at 20 Hz using the CSAT-3 sonic anemometer (Campbell Scientific Inc., Logan, Utah, USA)."

For the ERA5 calculation, "instantaneous" fluxes were used. This sentence "Obukhov Length is the characteristic length scale of the atmosphere and is calculated from ERA5 data using surface sensible heat and moisture fluxes." Has been changed to this: "Obukhov Length is the characteristic length scale of the atmosphere and is calculated from ERA5 data using instantaneous surface sensible heat and moisture fluxes". We directly followed the suggested data download and equations from ECMWF. To make it easier for readers to repeat this process, we have included a citation to this resource from ECMWF and added this text "fluxes based on the suggested calculation from the European Centre for Medium-Range Weather Forecasts (Gusti, 2024)" to line 142.

9. The phrase "Eq. (5.7c) from Stull (1988)" is difficult to follow. I suggest rephrasing it as: "... was calculated using Eq. (1) following Stull (1988)".

Line 136 has been updated from "Eq. (5.7c) from Stull (1988)" to "Eq. (1) following Stull (1988)", as suggested.

10. Line 146: Please provide the manufacturer name and model number instead of listing only a URL. The same request applies to other instruments mentioned in the manuscript, including the 3D sonic anemometer, ion chromatography system, and AirSentry.

Lines 146 to 148 were updated to the following: "Biweekly NH3 ambient air concentration was measured using Radiello passive diffusion samplers purchased from Sigma Aldrich. The Radiello sampling system includes a diffusive body (part number: RAD1201) and adsorbing cartridge (part number: RAD168)", to include part numbers and manufacturer name for the passive sampling. For the ion chromatography system in line 150, the text "using ion chromatography (IC)" has been updated to: "analyzed on a cation IC using a 20 mM methanesulfonic acid eluent (0.5 mL min-1) on a Dionex CS12A ion exchange column with a CSRS ULTRA II suppressor and Dionex conductivity detector(Li et al., 2016)." The 3D sonic anemometer used was a CSAT-3 3-D Sonic Anemometer from Campbell Scientific. This description has been added to the NEON data section.

The AirSentry model (AirSentry II) and manufacturer name (Particle Measuring Systems) are both listed in lines 156-157 with the text: "The instrument used was the AirSentry II Point-of-Use IMS from Particle Measuring Systems located in Boulder, CO." The text has been updated to the following: "The instrument used was the AirSentry II Point-of-Use IMS (Particle Measuring Systems, Niwot, CO)", to avoid confusion.

11. Line 151-152: There are several types of denuders (e.g., annular, multi-channel, honeycomb), but I am not familiar with the term "University Research Glassware Denuders". Is this a proper noun or brand name? Please clarify or revise how this is described in the manuscript. And I do not agree with that the passive sampler have a low bias based on your low data.

University Research Glassware is the company that made the annular denuders used in the cited paper. We have updated the text in line 151 from "University Research Glassware Denuders" to read "annular denuders", so it will be clearer to the reader. In this section, we are citing other works that have considered the biases present in passive sampling techniques. Due to the lack of data overlaps, we were not able to directly compare passive measurements to another collocated data source. We are using other works that indicate a low bias to assess what the impact of those low biases would be on simulated NH3 fluxes.

12. Line 157: "Boulder, CO" may refer only to a location and does not specify the manufacturer or model. The correct company name is "Particle Measuring Systems." Please revise accordingly. What does "1/4" means? And please correct the unit from "C" to "°C".

The manufacturer and company name is already given ("Particle Measuring Systems"). The model name is also given ("AirSentry II"). Boulder, CO, is the location of purchase. The text has been updated to the following: "The instrument used was the AirSentry II Point-of-Use IMS (Particle Measuring Systems, Niwot, CO)", to avoid confusion. Thank you for catching the ¼", this means 0.25 inches. The text has been updated from imperial (¼") to metric (0.635 cm). A degree symbol has been inserted.

13. Line 159: Did you perform any tests to assess potential NH3 losses within the sampling tube? Given that NH3 is highly soluble in water, it may be adsorbed onto the tube walls under high humidity conditions.

The efforts taken to avoid NH3 loss to the inlet are described in lines 159-161: "The sampling inlet was 0.635 cm Teflon tubing, heated to 40 °C to reduce NH3 loss to the sampling tube. Inlet length was kept as short as possible to further prevent NH3 loss." We did not directly test NH3 losses to humidity, but we would not expect condensation onto the inlet tubing walls given the coating and heat maintained. The average seasonal temperature and RH is now included in the text: "The seasonal mean temperatures (relative humidities) are as follows: winter (December, January, and February) mean is -3 °C (30%), spring (March, April, and May) mean is 2 °C (44%), the summer (June, July, and August) mean is 15 °C (49%), and the fall (September, October, and November) mean is 8 °C (37%)." Given the low temperatures, heating the inlet to 40 °C is much hotter than the ambient and the typical RH values are below 50% in all seasons.

14. Line: 168: Please specify "the effect of NH3 ..." on what? The sentence is currently ambiguous.

The line "To investigate the effect of NH3 (g) sampling time resolution" has been changed to "To investigate the effect of NH3 (g) sampling time resolution on simulated fluxes" to explain that we are trying to understand how NH3 sampling time resolution will affect simulated fluxes.

15. Line: 169: The expression "30-min frequency" is potentially misleading. Since AirSentry's time resolution is already expressed as measured by "30-sec frequency", the 30-minute value generated here should be rephrase.

The text was updated from "30-minute frequency" to "30-minute resolution."

16. Figure 2: The labels "2021" and "2022" on the x-axis only need to appear once at the beginning of each year. Instead, it would be more helpful for readers to better illustrate seasonal trends if the figure included month labels across the full period.

Every other month is marked in the current version of Fig. 2. We attempted to include a label for every month; however, this made it challenging to read the labels on the x-axis.

17. Line 180: If my understanding is correct, the data count would exceed 3,000 if values below the detection limit or those reported as zero are also included.

We have added more explanation in the AirSentry methods, as discussed above, to address the missing values.

18. Line 190: Please correct the formatting of "Fig (S1)" to "Fig. S1".

The text in line 190 has been updated from "Fig (S1)" to "Fig. S1".

19. Line 202: Please specify what type of wet deposition data were used.

The text in line 202 was updated from "Wet deposition" to "Weekly precipitation wet deposition", to better reflect the collection methods used by the NADP NTN.

20. Line 203: Please use the same format for latitude and longitude notation as used for other sites to ensure consistency throughout the manuscript.

Thank you for catching this. All latitudes and longitudes have been updated to this format "40.3639, -105.5810". The text in line 203 was changed from "40.3639°N, -105.5810°E" to "40.3639, -105.5810".

21. Line 210: The phrase "dry deposition is generated" sounds unnatural. Dry deposition is typically inferred or estimated, not "generated." Please revise this expression accordingly. The same applies to the use of "generation" in reference to Vd.

The word "generated" has been changed to "estimated" when referring to Vd values.

22. Line 214: The abbreviation "(Vd)" should be introduced at Line 211 when "deposition velocity" is first mentioned.

The abbreviation for deposition velocity, "(Vd)", was added to line 211 and removed from line 214.

23. Line 215: Why "dry deposition velocity" is being used at here instead of Vd.

In lines 214 through 216, the terms "deposition velocity" and "dry deposition velocity" has been replaced with  $V_{\rm d}$ .

24. Line 217: Although I understand this is based on previously studies, please note that the Vd of NH3 is not so easily defined. This may partially explain the discrepancy in annual deposition amount discussed at Line 365-368.

Yes, we are just basing this on a few previous studies. We agree with the referee that this is a simplified and not perfect definition for  $V_{d(NH3)}$ .

25. Line 219: The bidirectional flux model by Massad et al. (2010) is more complex than other models used in previous studies, and the calculation of the  $\chi$ z0 is difficult. This is why the authors are required to conduct various sensitivity analyses. It would be better to describe the advantages behind selecting this model.

Thank you for this feedback, we have added the following line: "This model was selected because it estimates both emissions and deposition of NH3, uses a compensation point framework to capture these complex dynamics, and takes into account rapidly changing micrometeorology." This should better explain to the reader why we have selected this model and increase their understanding.

26. Line 222: If I understand correctly, the authors appear to have misunderstood the framework of the model of Massad et al. (2010). In this model, the direction of total flux is determined by the difference between atmospheric concentration ( $\chi a$ ) and  $\chi z 0$ . The canopy compensation point ( $\chi c$ ) is an intermediate parameter for determining  $\chi z 0$ , not the determinant of direction of total flux. This distinction is important because the model differs conceptually from simpler models. Please revise the explanation to ensure accuracy.

Thank you for noting this. As shown in Eq. (17), the overall flux is determined by  $\chi_a$  and  $\chi_{z0}$ , as you correctly indicate here. The text has been updated to give the proper description.

This line (226) has been deleted: "Canopy compensation point depends on the stomata resistance, cuticle resistance, and stomata compensation point."

The following text: "The model determines if the flux will be negative (deposition) or positive (emission) based on the relationship between the atmospheric concentration ( $\chi_a$ ) at a

given reference height (z) and the canopy compensation point ( $\chi_c$ )." has been replaced with: "The model determines if the flux will be negative (deposition) or positive (emission) based on the relationship between the atmospheric concentration ( $\chi_a$ ) at a given reference height (z) and the compensation point ( $\chi_{z0}$ ) at a defined distance (d) above the roughness length ( $\chi_{z0}$ )."

27. Figure 3: The term "surface compensation point" is incorrect; as far as I know, this terminology is not used in previous literature.  $\chi z0$  is the compensation point at height of (d+z0). Please correct "Stomata" to "Stomatal", and "cuticle" to "cuticular". Also, "laminar" is no need for Rbg.

Yes, Massad et al. (2010) describe  $\chi_{z0}$  as the compensation point at height (d+z0). We called it the surface compensation point here to encompass all compensation points to deal with all surface relationships. In Massad et al. (2010), it is not given an intuitive term. We appreciate the reviewer's feedback that calling it the "surface compensation point" adds confusion. We have removed the instances of "surface compensation point" and replaced them with " $\chi_{z0}$ " throughout the manuscript.

"Stomata" and "cuticle" have been updated to "stomatal" and "cuticular", respectively.

The word "laminar" was removed from the description of  $R_{bg}$  in Fig. 3.

28. Line 235: Is Figure 3 showing "relationship" rather than a conceptual diagram? Please clarify and revise the description accordingly.

Good point, referring to Fig. 3 as a conceptual diagram is a better description. We have updated this sentence in line 235: "The relationship between resistances and compensation points is shown in Fig. 3" to: "A conceptual diagram of resistances and compensation points is shown in Fig. 3".

29. Line 239: Obukhov length, displacement and roughness length have already been defined as L, d, and z0.

The sentence including displacement height and roughness length has been updated to read: " $R_a$  was calculated according to Thom (1975), where z is 25.35 m, d is 7.15 m, and the roughness length is 1.65 m". The sentence now does not redefine the shorthand, but still includes the numeric values. On line 244, "Obukhov length" was replaced with "L".

30. Line 245: Please delete "captures the aerodynamic resistance from within the canopy layer and". While this may apply to Rac, it is incorrect in the context of Rg.

The phrase "captures the aerodynamic resistance from within the canopy layer" was deleted from line 245.

31. Line 247: The phrase "using Eq. 16 and Eq. 17 from equations of Massad et al. (2010)." may confuse readers, especially since the present manuscript also contains Eq.(16) and Eq.(17).

The equation numbers are included to help a reader reference where in Massad et al. (2010) they should look to calculate the  $\alpha$  parameter. We understand the referee's concern here. However, we have decided to keep the equation numbers for easier referencing to the original material.

32. Line 252: Could you elaborate on how the parameter Rbg was determined? This parameter are not widely compiled, and your approach would provide valuable information for future studies.

The Rbg parameter equation has been updated to more clearly follow the description in Nemitz et al. (2001). The following text has been added to our description to explain how the parameter was calculated (lines 271-273): "Ground boundary layer resistance ( $R_{bg}$ ) is based on Nemitz et al. (2001), where  $u_g$  is the wind speed at the ground, which we approximate as 5% of the wind speed at tower top (25 m), and  $z_l$  is the upper bound height of the logarithmic wind profile above the ground, which we approximate as 10% of the canopy height (Nemitz et al., 2001)"

33. Line 257: According to Table 1 in Zhang et al. (2003 https://doi.org/10.5194/acp-3-2067-2003), I could not find a rstmin value of "225" for any land use category. Is the mistake of 250?

This is not a mistake, we used a combination of the values for the two fauna types in RMNP, evergreen needleleaf trees and deciduous broadleaf trees + shrubs. Using land surface types from the NEON database, we assumed 75% evergreen needleaf trees (Rst min = 250) and 25% deciduous broadleaf trees + shrubs (Rst min = 150). This sentence: "The minimum value for  $R_{st}$  (225 s m-1) was determined using Table 1 of Zhang et al. (2003)" was updated to: "The minimum value for  $R_{st}$  (225 s m-1) was determined using Table 1 of Zhang et al. (2003), assuming 75% of the land surface was evergreen needleaf trees and 25% was deciduous broadleaf trees and shrubs."

34. Also, did you assign Rst at nighttime to be infinite value considering the stomatal closure following Zhang et al. (2003)? This is a critical assumption in modeling bidirectional exchange of NH3.

No, we used the equation as listed in Table 2 of Nemitz et al. (2001).

35. Line 258: Based on your response to Referee #1, it seems there may be a misunderstanding regarding Rw. As I understand it, this is an empirical formulation that accounts for four different vegetation types, not only for "Douglas Fir." Moreover, the effect of LAI and Temperature is considered at this empirical formula. It is important to avoid applying such models blindly without a full understanding of their basis.

We see how, in this case, including the phrase "predominantly Douglas Fir" has lead to confusion about how Rw was parameterized. Yes, the empirical formula includes LAI and temperature affects, but does not use the equation suggested by Referee #1. The text has been updated to "Cuticular resistance ( $R_w$ ) was calculated according to the proposed corrected parameterization as described in Massad et al. (2010), for a forest ecosystem." This is following equation 24 in Massad et al. (2010), using the parameter proposed for a forest ecosystem.

36. As an additional point, changing the cuticular resistance formulation can also significantly affect NH3 fluxes in addition to the emission potential. I recommend considering the case study of Xu et al. (2023 https://doi.org/10.1016/j.atmosenv.2023.120144) using a bidirectional exchange model in your uncertainty discussion or future sensitivity analysis.

For this work, we used the cuticular resistance described in equations 24 and 23 of Massad et al. (2010) directly. Xu et al. (2023) include this formula in their Table 2, as an option for calculating Rw. However, they found that the Rw formula from Sutton et al. (1998) was a better fit for their observations. This is interesting and may be important as we think about using our data for model parameterization efforts. However, for this paper, we use the Massad et al. (2010) equation as described. It will be interesting to see if this formula for Rw aligns with our flux measurements in the future.

- 37. Line 262: This is self-evident from Eq.(8), and Eq.(9) is unnecessary. Is there any case in which relative humidity exceeded 100% at the NEON site?
- Eq (9) is necessary here, because if RH exceeds 100% the calculated Rw is different between Eq (8) and Eq (9). It is theoretically possible to achieve RH values that exceed 100%, so we have included Eq (9) to be consistent with Massad et al. (2010).
  - 38. Line 263: Although the exclusion of HCl likely has minimal influence on the results, was there a specific reason it was not considered in the calculation? I suspect that the higher acidity ratio (AR) observed in winter may be due to extremely low NH3 concentrations. Is it possible that a higher AR facilitated NH3 deposition in the site? According to the study by Xu et al. (2023), Rw can be highly sensitive to this factor.

Rw is sensitive to AR, it and RH drive the Rw Eqs. (8/9). Figure 7 of Massad et al. (2010) shows the dependence of Rw on AR when RH is held constant. HCl was excluded from this analysis for two reasons: 1. HCl gas is not measured by CASTNET at the site, and 2. Previous measurements of HCl at RMNP have had very low concentrations.

39. Line 269: The stomatal and ground compensation points have already been defined as  $\chi s$  and  $\chi g$ .

The text "stomatal and ground compensation points" has been changed to " $\chi_{st}$ " in accordance with this comment and comment 43.

40. Line 270: Why did you not use the formula of Massad et al. (2010), which calculates  $\Gamma$ s for Un-managed site based on nitrogen input? The  $\Gamma$ s value calculated by this formula is about 10 times larger than the value used in this study. Do you expect this have any impact on the flux calculations?

We use a calculated value for the  $\Gamma_{st}$  because we made measurements of foliage at the site which allows us to directly calculate an estimated  $\Gamma_{st}$ . This value is different from values predicted by Massad et al. (2010). However, these emission potentials are highly variable from both Massad et al. (2010), and Zhang et al. (2010). This may have impacted flux calculations, and supports that increased measurements of foliage should be made to improve stomatal emission potential estimates.

41. Line 272: What exactly do you mean by "ratios" in this context?

Ratios here was just meant to indicate to the reader that they are unitless values. The text has been updated to "Emission potentials describe the potential for surface emission."

42. Line 275: Same comment with comment 39 for χs and χg. Also, "Eq. (3) and Eq. (5) of Stratton et al. (2018)" could confuse readers, as the equation numbers overlap with those in your manuscript.

"Ground compensation point" was replaced with " $\chi_g$ ". As noted above, we appreciate the referee's feedback and understand how this may be confusing. However, we have decided to leave these equation numbers to aid the reader in referencing the original source material.

43. Line 281: The authors described "ground compensation points were calculated according to Massad et al. (2010)" in Line 269. However, they also described soil compensation point was calculated according to Stratton et al. (2018) in Line 275, and  $\chi g$  is not calculated follow the form of Eq. (11). If this is the mistake of  $\Gamma g$ , I can understand. Which is correct?

Thank you for catching this. "Ground compensation point" should not be included in line 269. That section is describing the stomatal compensation point. Eq. (11) shows the calculation of  $\chi_{st}$ . Eq. (12) shows the calculation of " $\chi_g$ ", which follows the indicated equations from Stratton et al. (2018).

44. Line 286: Same comment with 40 for χc.

"Canopy compensation point" has been replaced with " $\chi_c$ ".

45. Line 287: χa was already defined and explained earlier in Line 222.

The line "where  $\chi_a$  is the atmospheric NH3 concentration" has been deleted.

46. Line 291: Same comment with 43 for d and z0. Also, note that it is conventional to use lowercase z, not uppercase Z. Please revise accordingly.

Line 291 has been updated to directly use "d" and " $z_0$ ". Thank you for catching the typo of " $Z_0$ " in this location.

47. Line 296: The explanation of inferred flux here is scientifically incorrect. It raises concerns about whether the authors even fully understand the resistance model framework. Also, "roughness height" is incorrect as already been pointed out by Referee #1.

Thank you for catching the accidental use of "roughness height" here. It would be helpful for the referee to explain what they take issue with in line 296. Line 296 read "Finally, the total flux was calculated following Eq. (17) (Massad et al., 2010)" at the time of this review. We have updated the text from: "Finally, the total flux was calculated following Eq. (17) (Massad et al., 2010). NH3 flux is defined in this framework as a difference between the roughness height compensation point and the NH3 concentration at that height, scaled by the aerodynamic resistance." To this: "Finally, the total flux was calculated following Eq. (17) (Massad et al., 2010). NH3 flux is calculated in this framework as a difference between the  $\chi_{20}$  and  $\chi_{a}$ , scaled by  $R_{a}$ ." We see how use of the term "that height" is ambiguous and may confuse the reader.

48. Line 304: Please correct the section title to "Simulated bidirectional exchange flux of NH3".

We changed the section title in line 304 from: "Simulated bidirectional exchange of NH3" to "Simulated bidirectional exchange fluxes of NH3".

49. Line 307-309: This explanation has already been presented in the Methods section. And I cannot agree using the word of "relative magnitudes". Again, "Surface compensation point" is not an appropriate term; please revise all instances in the

manuscript. I suggest adding a plot of the "difference" between  $\chi a$  and  $\chi z 0$  in Figure 4. This "difference" directly determines the direction and magnitude of the flux and would provide valuable insight into seasonal variation.

We use the term "relative magnitudes" to capture that both the sign and size of  $\chi_{z0}$  and  $\chi_a$  matter. You are correct, using equation 17, the difference between  $\chi_{z0}$  and  $\chi_a$  determines the direction and magnitude of the simulated NH3 flux. The sentence: "NH3 flux direction is determined by the relative magnitudes of the NH3 concentration and the surface compensation point (Fig. 4a.)." has been updated to "NH3 flux direction is determined by the difference between  $\chi_{z0}$  and  $\chi_a$  (Fig. 4a.)."

50. Line 313: For clarity, please define the seasons as used in your study (e.g., spring = March to May). From Figure 4, it is not clear how "The largest periods of net emission occur in the spring" was concluded. It appears that the largest emissions occur from late June to July. Why are large depositions observed before and after this period (even in same summer)? What factors do you think are influencing these patterns?

The following text has been added to describe the seasons "winter (December, January, and February)", "summer (June, July, and August)" and "spring (March, April, and May)". Thank you for catching the error about net emissions. The largest net emissions do occur in the summer. It is an interesting question, this may be driven by the variability in transport patterns of NH3 to the park, leading to highly variable NH3 concentrations.

51. Line 321, 323: Same comment with 28.

"stomata" and "cuticle" have been replaced with "stomatal" and "cuticular"

52. Line 324: Based on Figure 5, it is difficult to support the claim that "Winter periods of net emission (see Fig. 4b) are driven by the ground flux." If these fluxes exhibit seasonal variation, boxplots may obscure such information, a time series plot would be more appropriate to reveal these dynamics because total flux is sum of these fluxes. Furthermore, as previously mentioned, some models assume no soil emission under snow-covered conditions. Then, why is soil emission estimated to be larger in winter when temperatures are low, and snow is present?

Considering the boxplot shown in Fig. 5, only the ground fluxes have large emissions. The stomatal and cuticular fluxes have a maximum value very close to zero. Therefore, emissions must be driven by fluxes from the ground. For this work, we did not incorporate a different parameterization for soil-emissions under snow cover conditions because of the large uncertainties in the effects of snow on NH3 fluxes. This will be crucial future research to understand NH3 fluxes in areas that have snow cover.

53. Line 325: Do you have any hypotheses or supporting information on how snow cover affects the flux?

We have added a small investigation of the effects of snow cover, by setting ground compensation point equal to zero in the wintertime. Some previous works suggest that snow cover would shut off ground emissions. However, there is limited evidence of fluxes directly to/from the snow surface. It would be a great focus of future research to understand how NH3 is deposited to the snow surface.

54. Line 338: The term "surface exchange" may be more appropriate than "dry deposition" here.

We specifically use "dry deposition" because the discussion is around elevated NH3 concentrations leading to deposition fluxes.

55. Line 346: The phrasing should be revised to "we also observe peak deposition fluxes" for clarity.

The phrase "we also observe peak fluxes" has been changed to "we also observe peak deposition fluxes".

56. Figure 6: While the discussion of diurnal variation is valuable, analyzing the full annual dataset may obscure seasonal characteristics. Do the observed diurnal patterns hold true across all seasons, for example, during winter when concentrations and air temperature are lower? Are stronger daytime emissions observed during summer?

We previously looked at the diurnal pattern across seasons and found that the general shape of the diel profile was the same. Although the values differed, with the strongest daytime deposition observed in the summer and the strongest daytime observed in the winter.

57. Line 365-368: Is it also possible that the applied bidirectional flux model underestimates the actual amount of dry deposition? This relates to the previous question: why is dry deposition substantially higher in May and August compared to other months? Could the low dry deposition in June and July be due to large daytime emissions? Since wet deposition also become lower during these months, did less rainfall favored NH3 emission?

It is possible that the applied bidirectional model underestimates dry deposition. We will hopefully have a better understanding of this when we compare our gradient method fluxes of NH3 from measurements on the NEON tower. For the scope of this paper, we are unable to assess this directly. In the text we are careful to say that our annual value for NH3 deposition is smaller than previously estimated, but not indicate that this necessarily means that the bidirectional flux model is the correct answer.

58. Line 379: According to Figure 7(b), the largest NH3 dry deposition appear in "May" and August. Please revise the text accordingly.

Yes, thank you. The two maximum months are given as "May and August" in the text now.

59. Line 380-382: Can you explain why the proportion of reduced nitrogen is so high in this area, in relation to the transport and sources mentioned above?

This could be due to the reduction in emissions of oxidized nitrogen, or the large CAFO emissions of NH3 in the CO Front Range. In Lines 353 to 360 we discuss the emissions and transport of NH3 to the park.

60. Line 389: Please be consistent in terminology. Use either "time resolution" or "time-resolution" throughout the manuscript, but do not mix both styles.

Instances of "time resolution" have been replaced with "time-resolution".

61. Line 406: Please consider expanding your discussion on the reasons for the differences in modeled flux. If I understand correctly, one key factor may be the diurnal variation in the flux: At high temporal resolution, models usually reproduce large daytime NH3 emissions driven by increasing temperature. At night, stomatal closure and reduced turbulence lead to larger stomatal and in-canopy aerodynamic resistances, suppressing emissions from both stomata and ground. Simultaneously, elevated relative humidity and acidity ratio (AR) enhance deposition. Therefore, deposition at nighttime may largely contribute to the annual dry deposition in 30-min resolution. However, at lower temporal resolution, these diurnal dynamics are averaged out, potentially leading to overestimation of emission and underestimation of deposition.

Thank you for this contribution to the discussion. In RMNP, we observe the largest deposition fluxes during the day, largely driven by NH3 concentrations. This can be observed in Fig. 6c. You are correct that at lower temporal resolution, these diurnal dynamics are averaged out, which leads to an improper estimation of emissions and deposition. We have added this text: "Simulated NH3 fluxes have a strong diel pattern when simulated at 30-minute resolution (see Fig. 6c), due to changes in NH3 concentration and meteorology. These complex dynamics are averaged out when an average NH3 concentration is used, which leads to an underestimation in deposition." to line 415.

62. Figure 10: While the overall trends of deposition and emission appear roughly consistent, there are notable differences in the magnitude of deposition fluxes in some cases. Both fluxes seem to exhibit systematic bias. What could be causing this? It is also puzzling that the annual dry deposition totals are same despite these differences.

We are not sure what the referee means here by "both fluxes seem to exhibit systematic bias". As such, we have not made changes to the text.

63. Line 438: Does this mean that you applied a monthly diel pattern of each month? If so, this sentence is an insufficient explanation.

Yes. We have updated this sentence from: "Annual deposition from all flux simulations using a monthly diel pattern fell within 2% of the annual deposition using the annual average diel pattern." to "Annual deposition from all flux simulations using each different monthly diel pattern fell within 2% of the annual deposition using the annual average diel pattern." To indicate that the results were the same for the diel pattern of each month.

64. Line 443: Please revise "Dry deposition inferential" to "Bidirectional exchange".

"Dry deposition inferential" was replaced by "Bidirectional exchange."

65. Line 446: Again, how did you simulate 30-min fluxes using ERA5 reanalysis data with 1-hour temporal resolution?

We did this by allowing each hour to represent both 30-minute time steps it contained.

66. Line 451-452: This sentence is unclear and please rephrase. Figure 11 suggests that the ERA5-based flux show smaller emission compared to the NEON simulation, which could result in the higher dry deposition amount.

That is correct; however, the ERA5 flux simulations also have smaller deposition fluxes. When we consider the annual flux, both emission and deposition play a role.

- 67. Figure 11: The title appears redundant since it merely repeats the caption. Thank you, we have removed the title here.
  - 68. It may be more informative to plot by specifying season or by day/night. This could help identify reasons for the discrepancies more clearly.

We looked at the plots by season and by day/night. Ultimately, however, we felt this orientation best aided our discussion.

69. Line 460: Same comment with 44 for Ra and  $\chi$ z0.

The long form of each was replaced with the shorthand.

70. Line 472: Same comment with 66 for u\* and L.

The long form of each was replaced with the shorthand.

71. Line 480: Same comment with 68 for L.

If the u\* from ERA5 were corrected using NEON, would the simulation results become more consistent between the two datasets?

They become more consistent, but it does not fully account for the discrepancy. This is described in lines 501 to 505.

72. Line 481-482: Do the observed differences in Ra directly cause the discrepancies in dry deposition amount? If so, please describe. Also, it is unclear why χz0 values are consistent despite significant differences in Ra (in Line 461-462).

Yes, from our analysis, differences in  $R_a$  appear to drive differences in simulated NH3 fluxes. The  $\chi_{z0}$  values are likely consistent despite the difference in  $R_a$  because there are many factors contributing to its calculation.

73. Moreover, Figure S2(h.) shows a large discrepancy in RH (R2 = 0.34), and ERA5 values being substantially higher than NEON. As RH is a critical input for Rw, how does this discrepancy influence fluxes? Why was this not discussed in the main manuscript?

When we directly compared the  $R_{\rm w}$  results from simulations with NEON and ERA5, we did not find a large discrepancy between them.

74. Line 485: Considering the editor's comment, the phrase "best simulated" may be inappropriate at this stage.

Good point. We have updated the conclusions in accordance with this comment and the editor's comments. The first sentence now reads "Fluxes of NH3 (g) can be simulated using a bidirectional model".

75. Line 489: As noted previously, please use consistent terminology throughout the manuscript: either "bidirectional" or "bi-directional," but not both.

Use of "bi-directional" in the conclusion instead of "bidirectional" has been updated to be consistent with the rest of the document.

76. Line 510-514: This methodology is quite promising. If you could briefly describe its potential applicability to other sites or its utility for future research, this would enhance the academic contribution of this study.

Thank you for contributing to shaping this section in a way that will aid future research efforts. This sentence has been added: "Understanding how to correct biases introduced through the use of reanalysis data would allow improved modelling of NH3 bidirectional fluxes in regions lacking high-time resolution measurements."

77. Figure S7: The meaning of "calculated fractional differences" is unclear. What can be seen from this figure is that the magnitude of flux in response to the χa scaling factor differ considerably by season, regardless of the two time periods.

We see where the confusion came from here and appreciate you pointing it out. We have changed the phrase "calculated fractional differences" to "relative changes". We specifically looked at the relative changes to understand how the flux magnitudes are changing given different initial fluxes.